# Optimization of the dosage of chopped basalt fibers in asphalt pavement surface course materials for semi-rigid base with functional requirements

Xiangbing Xie[1]*, Yahui He[1], Chenchen Liu[1], Kaiwei Wang[1], Zhezhe Fan[2]*, Huixia Li[3], Jinggan Shao[4]

1 School of Civil Engineering and Architecture, Zhengzhou University of Aeronautics, Zhengzhou, Henan, China, 2 School of Civil and Transportation Engineering, Henan University of Urban Construction, Pingdingshan, Henan, China, 3 School of Civil Engineering, Fujian University of Technology, Fuzhou, Fujian, China, 4 Henan College of Transportation Engineering Technology Group Co., Ltd., Green High-performance Material Application Technology Transportation Industry R&D Center, Zhengzhou, Henan, China

* xiexiangbing.good@163.com (XX); fanzhezhe0610@163.com (ZF)

**Data Availability Statement:** The datasets generated and analyzed during the current study are available in the Zenodo repository (DOI: 10.5281/zenodo.12701618).

## Abstract

How to select suitable pavement materials for asphalt pavements according to the functional requirements of layers is still the focus of research by scholars in various countries. However, their effectiveness in combating high-temperature rutting and fatigue cracking in middle and lower layers is limited. To address this issue, a study optimized the incorporation of basalt fibers in different layers to improve road performance based on design specifications. Nine asphalt pavement structures with varying amounts of basalt fibers were assessed using an orthogonal test method. The optimal structure was determined considering factors such as fatigue life and overloading using the finite element method for modeling. Results showed that fiber dosage had a minimal impact on road surface bending subsidence and the location of tensile strain in the lower layer. Shear stresses were concentrated mainly at the outer edges of loads. Optimal dosages of basalt fiber were determined for different layers: 0.3% for the upper layer, 0.1% for the middle layer, and 0.3% for the lower layer. The optimal structure consists of a strong base with a thin-surfaced semi-rigid base layer, with 0.3% for the upper layer and 0.1% for the middle layer. This study provided valuable insights into designing basalt fiber asphalt pavement structures.

## 1. Introduction

Asphalt concrete pavement, known for its comfortable driving experience, low noise levels, and easy maintenance, has been widely used in highway construction. Asphalt mixtures serve as the primary structural layer to bear pressure. It is essential to study their roles in different layers to comprehensively understand their performance. Additionally, in practical applications, various environmental factors and vehicle loads lead to asphalt pavement distress, such

**Funding:** The authors appreciate the financial support from the Youth Research Funds Plan of Zhengzhou University of Aeronautics (Grant No. 23HQN01007), Research on Application Technology and Equipment of Sprayed Basalt Fiber Reinforced Concrete (Grant No. 2020J-2-12), Research on Key Technologies of Application of Basalt Fiber and its Products in Highway Engineering (Grant No. 2021J5), the National Natural Science Foundation of China (Grant No. 51378474), Fund of Leading Talent in Science and Technology Innovation (Grant No. 194200510015), Science and Technology Department of Henan Province (Grant No. 192102210047), Scientific Research and Development Project of Zhengzhou Lutong Highway Construction Co., Ltd. (Grant No. 2021JK-11) and Natural Science Foundation of Henan Province (Grant No. 242300421258).

**Competing interests:** The authors declare no conflicts of interest.

as ruts, potholes, and cracks [1, 2], significantly impacting the lifespan and performance of the asphalt pavement [3].

To address the poor performance of mixtures, road builders actively use various additives for enhancement [4–6]. Among the various additives, fibers such as lignin fibers, basalt fibers, steel fibers [7], carbon fibers [8], glass fibers [9], and polyester fibers [3] have been acknowledged for their effective enhancement of the performance of asphalt mixtures. However, basalt fibers, as an environmentally friendly fiber material with excellent mechanical properties, good surface wettability, and a wide working range, are widely used as reinforcing materials in various fields [10–12]. Meanwhile, basalt fibers are added to asphalt mixtures to form composite materials, which can improve the road performance of asphalt mixtures, especially enhancing the high-temperature stability [13, 14], low temperature cracking resistance [15, 16] and fatigue resistance of asphalt mixtures [14, 17].

Today, with the increase in traffic volume, basalt fiber asphalt mixture is considered an effective material for enhancing road performance. This includes improving fatigue resistance, resistance to water damage, and crack resistance [18]. For example, the optimal mixing amount of basalt fiber in stone mastic asphalt (SMA) concrete is 0.35%. Its dynamic stability and Marshall stability performance are superior to those of lignin fibers and polyester fibers, particularly excelling in high-temperature stability [19, 20]. Using the asphalt concrete (AC-13C) mixture, different quantities of basalt fibers (0.1%, 0.2%, 0.25%, 0.3%, 0.35%) were incorporated, and high-temperature rutting tests were performed. The results indicate that the optimal fiber content is 0.25% for achieving the best road performance [21]. For the AC-13C asphalt mixture, the optimal basalt fiber content is 0.3%, leading to an approximately 35% improvement in dynamic stability at the optimal dosage [22, 23]. The overall performance of fiber asphalt mixtures improved when basalt fibers, lignin fibers, and polyester fibers were added to AC-20 asphalt mixtures at 0.3%, 0.4%, and 0.2%, respectively [24]. Basalt fiber effectively enhances the low-temperature performance of asphalt mixtures, making them more suitable for low-temperature environments. According to the low-temperature performance, the optimal basalt fiber content for AC-20 mixtures is 0.3%. At this level, the network structure of basalt fibers enhances the integrity of asphalt mixtures and effectively retards the expansion of microcracks [25, 26]. To investigate the role of basalt fibers in enhancing the road performance of asphalt mixtures, it was found that the optimal dosage of basalt fibers was 0.3%, and the optimal asphalt dosage was 4.63% [27]. After adding 0.3% basalt fibers to asphalt mixtures, the dynamic stability and maximum destructive strain of the asphalt mixtures improved, leading to optimal overall road performance [28]. The high and low-temperature stability and water resistance properties of asphalt mixtures were significantly enhanced by incorporating basalt fibers, with 0.3% identified as the optimal dosage [29, 30]. Basalt fiber is the most suitable type of fiber with an optimal length of 6 mm. The addition of basalt fiber significantly enhances engineering properties, including high temperature stability, low temperature crack resistance, and water sensitivity. The recommended basalt fiber content is 0.4 wt% [31]. Asphalt mixtures containing 0.3% basalt fibers, 4 mm in length, showed the highest fracture toughness compared to other mixtures [32].

In the design optimization of pavement structures, the asphalt surface layer is positioned differently within the road structure, experiencing various stresses. The upper layer of the road surface bears the surface layer's function while also coping with low-temperature contraction stress. In material design, it is necessary to consider low-temperature anti-cracking performance. The middle surface layer must withstand and distribute the stress transferred from the upper layer. It is prone to shear damage, so material design needs to focus on high-temperature rutting resistance. Below the surface layer, the bearing and distribution of stress transfer occur simultaneously. This layer must also withstand bending and tensile stress from traffic

loads, making it susceptible to initial fatigue damage. Therefore, special attention must be paid to the material design to enhance its resistance to fatigue cracking [33]. The structural mechanical analysis model of asphalt pavement was established by analyzing the early damage mechanism of asphalt pavement with a semi-rigid base layer. It is found that the internal tensile stress of asphalt pavement gradually increases and reaches the maximum value at the bottom, which determines the fatigue resistance of the pavement. The shear stress reaches the maximum value in the middle of the pavement, influencing the rutting resistance. Additionally, the low-temperature shrinkage stress peaks at the top of the pavement, affecting the resistance to low-temperature shrinkage cracking [34–36]. Employing fiber optic grating testing technology, a three-dimensional dynamic strain response analysis of asphalt pavement under typical truck loads was performed. The analysis revealed a significant compressive stress impact on the upper and middle layers, while the lower layer exhibited a cyclic "compression-tension" stress variation, making it more susceptible to fatigue cracking [37]. Structural analysis using ANSYS finite element software revealed that the middle layer of the asphalt pavement mainly carries the shear stress in the rut resistance zone, while the lower layer primarily resists the tensile stress caused by the load on the asphalt pavement, highlighting its importance in fatigue crack resistance [38]. Consequently, the functional requirements of asphalt pavement in China include resistance to surface cracking, skid resistance, thermal shrinkage stress, and water damage in the surface layer. The middle layer is designed for high-temperature stability, while the lower layer is engineered for fatigue crack resistance and high-temperature stability.

Ren et al. utilized a virtual testing method to investigate the low-temperature fracture characteristics of asphalt mixtures. Firstly, the loading mode of the SCB indoor test was determined, and the fracture characteristics under different loading modes and temperatures were analyzed. The influence of the fine structure of asphalt mixtures on their fracture properties, as well as the laws governing the influence of the structural properties and components of aggregates on these properties, were investigated using virtual testing methods. Finally, the asphalt mixture composition was optimized, and the results were verified through conventional indoor tests [39]. The structural model of cement-stabilized aggregates in semi-flexible pavement was constructed using the theory of a laminated elastic system. Orthogonal experimental methods were utilized to study the effects of different parameter combinations on the pavement structure. This included investigating the modulus and thickness of the surface layer, the sublayer, the base layer, and the subbase layer on the tensile stress and permanent deformation at the bottom of the layer. By analyzing the key indexes and considering the mechanical properties and economy, a preferred pavement structure combination system is proposed [40].

Basalt fibers can improve the road performance of asphalt mixtures, but less research has been done on the layer function aspect of basalt fiber asphalt pavement structures. In this paper, basalt fiber asphalt pavement structure is the research focus. An orthogonal test method was used to vary the basalt fiber dosage in nine different combinations of asphalt pavement structures. The study utilized finite element method to create a three-dimensional finite element model of the asphalt pavement structure under various mobile load combinations. The evaluation criteria included bending subsidence of the road surface, tensile strain at the bottom layer underneath, and shear stress at the bottom-middle surface layer. Factors such as fatigue life and overloading were considered to determine the optimal pavement structure. This research establishes a solid theoretical foundation for enhancing pavement performance, extending service life, and decreasing maintenance costs.

**Table 1. Performance indicators of modified asphalt.**

| Parameters | Requirements | Experimental results | Experimental method |
|---|---|---|---|
| Penetration (25%)/0.1 mm | 60–80 | 71 | GB/T 4509 |
| Softening point/˚C | ≥ 60 | 79 | GB/T 4507 |
| Ductility (5˚C,5 cm/min)/cm | ≥ 30 | 34 | GB/T 4508 |
| Penetration index PI | -1.5–1 | 0.16 | |
| Solubility/% | ≥ 99.0 | 99.5 | GB/T 11148 |
| Flashpoint/˚C | ≥ 230 | 281 | GB/T267 |

## 2. Materials and methods

### 2.1 Raw materials and mix design

**2.1.1 Asphalt.** In this study, SBS I-C modified road petroleum asphalt from Fuwei Chemical Co. was used, and its performance indexes are shown in Table 1 below.

**2.1.2 Fiber.** Chopped 6mm/17 μm basalt fiber was used in this experiment, and its various performance indicators are shown in Table 2.

**2.1.3 Aggregate.** The high-quality limestone from the material yard of Hongtu Highway in Inner Mongolia is used, and the mineral powder is calcium carbonate powder. According to the requirements of ' Highway Engineering Aggregate Test Procedure ' (JTG E42-2005), the performance indexes of different coarse and fine aggregates were tested. The test results are shown in Tables 3–5.

**2.1.4 mix design.** According to the specification and the test results of raw materials, the gradation design curve is fitted, and the aggregate gradation of SMA-16, AC-20 and ATB-30 is determined. The gradation is as shown in Tables 6–8.

**2.1.5 Determination of optimum oil-stone ratio.** The mixture ratio design of SMA-16 and AC-20 asphalt mixture adopts Marshall test method. The two sides of the specimen are compacted 75 times, and the oil-stone ratio is 4%, 4.5%, 5.0%, 5.5% and 5.0% respectively. The mix design of asphalt mixture of ATB-25 adopts Marshall test method. The double sides of the specimen are compacted 75 times, and the oil-stone ratio is 3.0%, 3.3%, 3.6%, 3.9% and 4.2% respectively. The vol-ume performance and Marshall test index of the specimen under the design compaction times were measured, so as to determine the best oil-stone ratio according to the design standard. The volume characteristics of asphalt mixture under each asphalt-aggregate ratio are shown in Tables 9–11.

According to the Marshall test results, as shown in Tables 9–11, the optimum asphalt-aggregate ratio of SMA-16 asphalt mixture is 5.3%, the optimum asphalt-aggregate ratio of AC-20 asphalt mixture is 4.7%, and the optimum asphalt-aggregate ratio of ATB-25 asphalt mixture is 3.8%.

**Table 2. Performance indexes of chopped basalt fiber.**

| Parameters | Requirements | Experimental results |
|---|---|---|
| Breaking strength/MPa | ≥ 1200 | 1260 |
| Elongation at break/% | ≤ 3.1 | 2.65 |
| Oil absorption/% | ≥ 50 | 53 |
| Flammability content% | 0.1–1.0 | 0.5 |
| Moisture content% | ≤ 0.2 | 0.14 |
| Flammability | open flame otherwise | open flame otherwise |

**Table 3. Technical index of coarse aggregate.**

| Inspection project | Unit | Coarse aggregate test results | | | Specification requirement | Test method |
|---|---|---|---|---|---|---|
| | | 20~40mm | 10~20mm | 5~10mm | | |
| Aggregate crushing value | % | 15.5 | —— | —— | $\leq$28 | T0316-2005 |
| Aggregate abrasion loss | % | | 10.6 | | $\leq$30 | T0317-2005 |
| Apparent relative density of aggregates | —— | 2.752 | 2.731 | 2.787 | $\leq$2.5 | |
| Relative density of aggregate gross volume | —— | 2.736 | 2.702 | 2.737 | Measured value | T0304-2005 |
| Aggregate water absorption | % | 1.12 | 1.65 | 1.61 | $\leq$3.0 | |
| Firmness | % | 6.8 | —— | —— | $\leq$12 | T0314-2005 |
| Flat elongated particles content | % | 16 5.5 | 15 | 15 | Measured value | T0312-2005 |
| | % | 5.5 | 5.1 | 5.0 | Required value | |
| Particle content less than 0.075 mm (water washing method) | % | 0.12 | 0.31 | 0.43 | $\leq$1 | T0310-2005 |

## 2.2 Selection of influencing factors

Basalt fiber reinforced the gap-graded asphalt concrete SMA-16(BFSMA-16: Basalt Fiber Stone Mastic Asphalt 16), basalt fiber reinforced the dense gradation asphalt concrete AC-20 (BFAC-20:Basalt Fiber Asphalt Concrete 20), and basalt fiber reinforced asphalt-treated permeable base ATB-25(BFATB-25: Basalt Fiber Asphalt-treated Base 25) were selected as factors according to research results from the literature [19, 24, 25, 28, 29], the dosage of BFSMA-16 was 0.1%, 0.2%, or 0.3%, the dosage of BFAC-20 was 0.1%, 0.2% %, or 0.3%, and the BFATB-25 content was 0.1%, 0.2%, or 0.3%. The level number was selected according to the principle that the number of factors $\leq$ the number listed in the orthogonal table, and the level number of the factor was consistent with the level number corresponding to the orthogonal table, which was selected as $L_9(3^4)$. Research on optimal pavement structure was conducted, see Table 12 for the test scheme [41].

## 2.3 Test method

**2.3.1 Dynamic modulus test.** The stress process experienced by the asphalt pavement surface layer under traffic loads is highlighted, emphasizing the use of a dynamic loading model for a more accurate analysis of the asphalt pavement's viscoelastic properties and stress

**Table 4. Test results of fine aggregate.**

| Inspection project | Unit | Test results of fine aggregate | | | |
|---|---|---|---|---|---|
| | | 3~5 | Stone chips | Specification requirement | Test method |
| Apparent specific gravity | —— | 2.726 | 2.712 | $\leq$22.5 | T0304-2005 |
| Mud content ($<$ 0.075 mm content) | % | 1.1 | 2.4 | $\leq$3 | T0333-2005 |

**Table 5. Test results of mineral powder.**

| Inspection project | | Unit | Test results | Specification requirement | Test method |
|---|---|---|---|---|---|
| Apparent specific gravity | | t/m$^3$ | 2.713 | $\leq$2.5 | T0352-2005 |
| Particle size range | $<$0.6 | % | 99.84 | 100 | T0351-2005 |
| | $<$0.15 | % | 91.91 | 90~100 | |
| | $<$0.075 | % | 71.93 | 75~100 | |
| Hydrophilic coefficient of mineral powder | | —— | 0.8 | $<$1.0 | T0353-2005 |
| Plasticity index | | —— | 3.7 | $<$4 | T0354-2005 |

**Table 6. Aggregate gradation of SMA-16.**

| Particle size/mm | The mass percentage passing through the following sieves (%) | | | | | | | | | | |
|---|---|---|---|---|---|---|---|---|---|---|---|
| | 19 | 16 | 13.2 | 9.5 | 4.75 | 2.36 | 1.18 | 0.6 | 0.3 | 0.15 | 0.075 |
| Upper gradation limit/% | 100 | 100 | 85 | 65 | 32 | 24 | 22 | 18 | 15 | 14 | 12 |
| Lower gradation limit/% | 100 | 90 | 65 | 45 | 20 | 15 | 14 | 12 | 10 | 9 | 8 |
| Gradation median /% | 100 | 95 | 75 | 55 | 26 | 19.5 | 18 | 15 | 12.5 | 11.5 | 10 |
| Synthetic gradation/% | 100 | 95.6 | 74.2 | 53.6 | 24.5 | 18.5 | 17 | 13.2 | 11.5 | 10 | 10 |

conditions. In the NCHRP program, the Simple Performance Tester (SPT) is used to determine the dynamic modulus of asphalt mixtures under uniaxial compression. This allows for the evaluation of material characteristics and serves as a foundation for designing asphalt pavement materials and structures.

According to Chinese national standards, the mineral gradation and optimal asphalt-to-aggregate ratio for SMA-16, AC-20, and ATB-25 mixtures are determined. The content of basalt fiber is then varied at four levels: 0%, 0.1%, 0.2%, and 0.3%. To measure the dynamic modulus, cylindrical specimens are prepared using a gyratory compactor, with metal pins attached to the specimens. The specimens are then placed in a constant temperature chamber for insulation. Finally, measurements are made by a test program to obtain the appropriate data.

**2.3.2 Establishment of the 3D finite element model.** The selected semi-rigid base asphalt pavement structure form is shown in Fig 1 [42, 43] traditional pavement structure, which represents the traditional pavement structure for highways. The viscoelastic characteristics of the upper layer SMA-16, middle layer AC-20, and lower layer ATB-25 are represented using the viscoelastic model in the finite element method.

Asphalt mixtures are viscoelastic materials, and their mechanical behavior is significantly dependent on time. The Burgers four-element mechanical model elucidates the viscoelastic properties of mixtures. When deformation is applied for a very short period, the mixture exhibits significant viscosity; however, the viscosity decreases when the same deformation is applied for a longer period. The creep and stress relaxation properties exhibited by the mixes over the experimental time range caused the mechanical modulus of the mixes to decrease

**Table 7. Aggregate gradation of AC-20.**

| Particle size/mm | The mass percentage passing through the following sieves (%) | | | | | | | | | | | |
|---|---|---|---|---|---|---|---|---|---|---|---|---|
| | 26.5 | 19 | 16 | 13.2 | 9.5 | 4.75 | 2.36 | 1.18 | 0.6 | 0.3 | 0.15 | 0.075 |
| Upper gradation limit/% | 100 | 100 | 92 | 80 | 72 | 56 | 44 | 33 | 24 | 17 | 13 | 7 |
| Lower gradation limit/% | 100 | 90 | 78 | 62 | 50 | 26 | 16 | 12 | 8 | 5 | 4 | 3 |
| Gradation median/% | 100 | 95 | 85 | 71 | 61 | 41 | 30 | 22.5 | 16 | 11 | 8.5 | 5 |
| Synthetic gradation/% | 100 | 94 | 84 | 73 | 58 | 41 | 29 | 22 | 15 | 12 | 8 | 4 |

**Table 8. Aggregate gradation of ATB-25.**

| Particle size/mm | The mass percentage passing through the following sieves (%) | | | | | | | | | | | |
|---|---|---|---|---|---|---|---|---|---|---|---|---|
| | 31.5 | 26.5 | 19 | 16 | 13.2 | 9.5 | 4.75 | 2.36 | 1.18 | 0.6 | 0.3 | 0.15 | 0.075 |
| Upper gradation limit/% | 100 | 100 | 80 | 68 | 62 | 52 | 40 | 32 | 25 | 18 | 14 | 10 | 6 |
| Lower gradation limit/% | 100 | 90 | 60 | 48 | 42 | 32 | 20 | 15 | 10 | 8 | 5 | 3 | 2 |
| Gradation median/% | 100 | 95 | 70 | 58 | 52 | 42 | 30 | 23.5 | 17.5 | 13 | 9.5 | 6.5 | 4 |
| Synthetic gradation/% | 100 | 96.5 | 69.2 | 57.5 | 50.3 | 43.5 | 32 | 22.8 | 15 | 11.5 | 9 | 6.5 | 5.2 |

**Table 9. Volumetric properties of SMA-16 mixes with different oil/gravel ratios.**

| Oil-rock ratio (%) | Gross bulk density (g/cm³) | Void ratio(%) | Mineral Gap Ratio(%) | Asphalt saturation (%) | Stability (kN) | Flow value (mm) |
|---|---|---|---|---|---|---|
| | | | Marshall double-sided each hit 75 times | | | |
| 4.8 | 2.459 | 4.1 | 14.5 | 70.3 | 9.33 | 37.26 |
| 5.1 | 2.461 | 3.8 | 15.4 | 72.5 | 10.27 | 35.32 |
| 5.4 | 2.466 | 3.3 | 15.6 | 77.6 | 8.06 | 35.65 |
| 5.7 | 2.458 | 3.2 | 16.1 | 81.3 | 8.43 | 31.21 |
| 6.0 | 2.462 | 3.0 | 16.5 | 81.6 | 8.65 | 28.45 |

**Table 10. Volumetric properties of AC-20 mixes with different oil/gravel ratios.**

| Oil-rock ratio (%) | Gross bulk density (g/cm³) | Void ratio(%) | Mineral Gap Ratio(%) | Asphalt saturation (%) | Stability (kN) | Flow value (mm) |
|---|---|---|---|---|---|---|
| | | | Marshall double-sided each hit 75 times | | | |
| 4.1 | 2.482 | 6.4 | 16.3 | 59.5 | 12.06 | 25.0 |
| 4.4 | 2.505 | 5.3 | 15.9 | 65.4 | 13.55 | 26.8 |
| 4.7 | 2.512 | 4.6 | 15.8 | 70.5 | 13.31 | 20.2 |
| 5.0 | 2.514 | 4.4 | 14.5 | 71.6 | 13.46 | 23.6 |
| 5.3 | 2.521 | 4.1 | 14.6 | 73.4 | 13.65 | 22.5 |

with time. The generalized Pliny series model, commonly used in the viscoelastic constitutive relationship of asphalt mixtures, succinctly reflects the creep and stress relaxation properties of the mixes.

In the finite element method model, the viscoelastic model utilizes the shear modulus of the Prony series to depict the material's time-dependent behavior. The input parameters include a set of Prony series coefficients and their corresponding relaxation times. The shear modulus can be calculated from the relaxed modulus using Eq (1), which is expressed as follows:

$$G(t) = \frac{E(t)}{2(1+\mu)} \tag{1}$$

Formula: G(t)−shear modulus; E(t)−relaxed modulus; μ−Poisson's ratio.

The shear modulus ratio obtained from the instantaneous shear modulus GT and normalized shear modulus GT as experimental data for fitting the Prony series in the finite element method is represented as g(t).

$$g(t) = 1 - \sum_{i=1}^{N} g_i (1 - e^{-t/\tau_i}) \tag{2}$$

**Table 11. Volumetric properties of ATB-25 mixes with different oil/gravel ratios.**

| Oil-rock ratio (%) | gross bulk density (g/cm³) | void ratio(%) | Mineral Gap Ratio(%) | Asphalt saturation (%) | Stability (kN) | Flow value (mm) |
|---|---|---|---|---|---|---|
| | | | Marshall double-sided each hit 75 times | | | |
| 3.1 | 2.354 | 6.8 | 15.2 | 54.4 | 6.55 | 1.44 |
| 3.4 | 2.370 | 5.7 | 15.1 | 60.3 | 7.64 | 2.54 |
| 3.7 | 2.384 | 4.8 | 14.6 | 66.4 | 8.38 | 3.10 |
| 4.0 | 2.382 | 4.6 | 14.8 | 69.6 | 8.69 | 3.14 |
| 4.3 | 2.383 | 4.0 | 15.1 | 73.1 | 8.25 | 3.16 |

Table 12. Orthogonal experimental design scheme.

| Test | Influencing factors | | | |
|---|---|---|---|---|
| | BFSMA-16(A) /% | BFAC-20(B) /% | BFATB-25(C)/ % | Blank (D) |
| 1 | 0.1 | 0.1 | 0.1 | 1 |
| 2 | 0.1 | 0.2 | 0.2 | 2 |
| 3 | 0.1 | 0.3 | 0.3 | 3 |
| 4 | 0.2 | 0.1 | 0.2 | 3 |
| 5 | 0.2 | 0.2 | 0.3 | 1 |
| 6 | 0.2 | 0.3 | 0.1 | 2 |
| 7 | 0.3 | 0.1 | 0.3 | 2 |
| 8 | 0.3 | 0.2 | 0.1 | 3 |
| 9 | 0.3 | 0.3 | 0.2 | 1 |

Formula: N–The number of terms in the Prony series;–Prony level parameters;–latency.

The broad-sense Prony series and the generalized Kelvin model share the same analytical form. The key distinction is that the former is not limited by the number and interval of terms, providing flexibility for experimental results to be freely specified. A set of delay times encompassing the entire experimental process is chosen. The parameters of the Prony series in Eq (2) are obtained using the nonlinear least squares curve fitting method.

Generally, when the Prony level is set to 5 terms, the measured shear modulus ratio can be fitted within 0.01 of the mean squared error. This allows for a more accurate simulation of the use of 5 Prony levels for parameter fitting in three types of asphalt mixtures. The fitting results of the Prony levels for the three mixtures are shown in Tables 13–15. The WLF parameters are

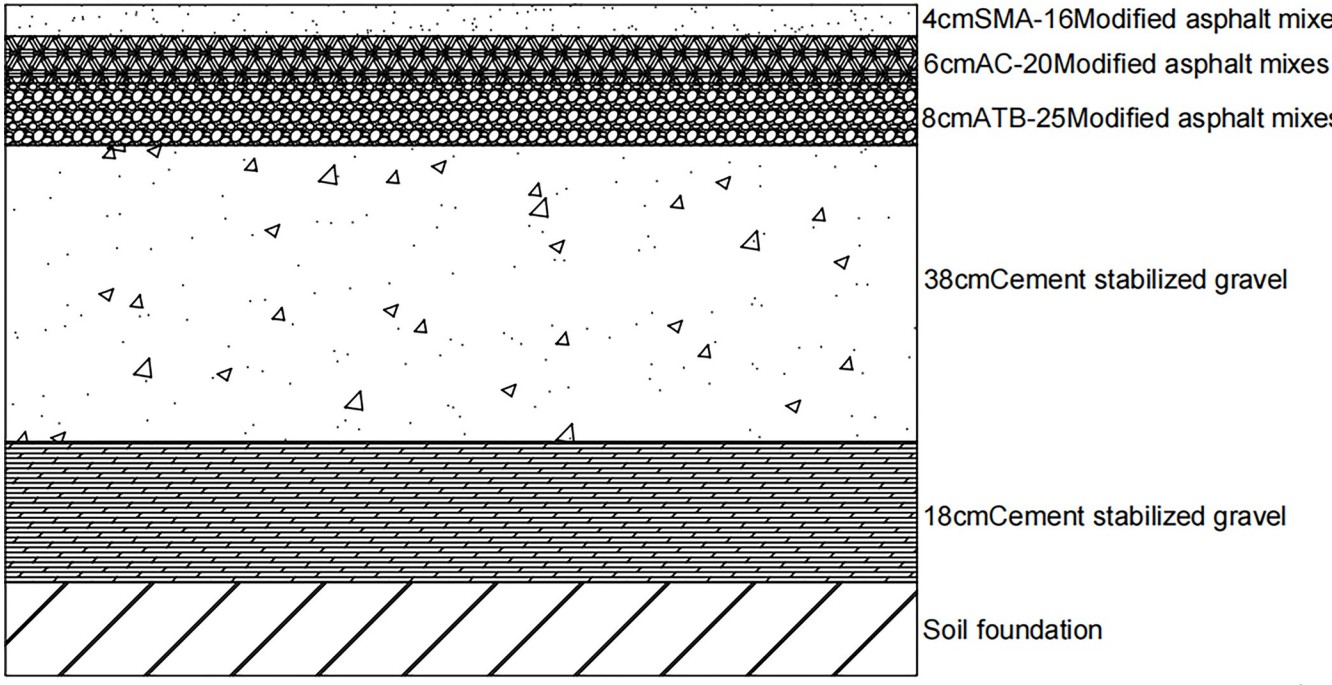

Fig 1. Traditional pavement structure.

**Table 13. Prony series parameters for SMA-16.**

| Parameters | Values | Parameters | Values |
|---|---|---|---|
| g1 | 0.4368 | τ1 | 1.26E-05 |
| g2 | 0.2742 | τ2 | 3.96E-04 |
| g3 | 0.1921 | τ3 | 6.12E-03 |
| g4 | 0.1023 | τ4 | 0.1341 |
| g5 | 0.0562 | τ5 | 12.875 |

**Table 14. Prony series parameters for AC-20.**

| Parameters | Values | Parameters | Values |
|---|---|---|---|
| g1 | 0.3342 | τ1 | 1.05E-05 |
| g2 | 0.2931 | τ2 | 1.51E-04 |
| g3 | 0.2723 | τ3 | 3.50E-03 |
| g4 | 0.0814 | τ4 | 0.1216 |
| g5 | 0.0345 | τ5 | 6.8132 |

presented in Table 16. The compressive modulus of the base layer was 7500 MPa, and the compressive modulus of the subgrade was also 10000 MPa.

The moving load adopted the standard axle load used in the current structural design of asphalt pavement in China, and the load adopted the double-circle vertical uniform load, that is, the standard axle load of 100 kN for a single axle load of the double-wheel set, the ground pressure of the tire was 0.7 MPa, the radius of action of the load was r = 0.1065 m, and the distance between the centers of the double circles was 3r = 0.3195 m. The X-direction was the driving direction, the Y-direction was the road cross-sectional direction, and the Z-direction was the road depth direction; the calculation points were selected according to the corresponding provisions in JTG D50-2017 "Code for Design of Highway Asphalt Pavement", as shown in Fig 2 below.

The three-dimensional finite element model was a five-layer structure of asphalt pavement, and the total thickness of the pavement was 74 cm. In the upper surface of the pavement role of the traffic load, the model length, width, and height were 6 m, 6 m, and 3 m, respectively. The mesh was divided into 0.1 × 0.1, and the load area coded division was 0. 05 × 0.05, and the calculation unit for the 8-node hexahedral reduction integral unit (C3D8R), the model and mesh division are shown in Fig 3 [44]. The model and mesh division are shown in Fig 6, where the X-direction is the direction of travel of the pavement structure, and the Y-direction is the depth direction of the pavement structure.

To simplify the calculation, a standard axle load of 0.7 MPa was chosen. The specific load locations and boundary conditions are shown in Fig 4.

**Table 15. Prony series parameters for ATB-25.**

| Parameters | Values | Parameters | Values |
|---|---|---|---|
| g1 | 0.3911 | τ1 | 2.71E-05 |
| g2 | 0.2503 | τ2 | 6.69E-04 |
| g3 | 0.2421 | τ3 | 9.54E-03 |
| g4 | 0.1132 | τ4 | 0.2013 |
| g5 | 0.0325 | τ5 | 10.893 |

**Table 16. WLF equation coefficients.**

| Parameters | $C_1$ | $C_2$ | R |
|---|---|---|---|
| SMA-16Values | 9.735 | 122.624 | 0.99912 |
| AC-20Values | 12.646 | 133.965 | 0.99821 |
| ATB-25Values | 18.423 | 162.879 | 0.99142 |

## 3. Test results and analysis

### 3.1 Dynamic modulus

Taking the SMA-16 test as an example, the relevant experimental test conditions and results are as follows:

1. Temperature: The environmental chamber that accompanies the SPT tester controls a temperature range from 5°C to 50°C. Considering the research objectives of this study, the test temperatures of 5°C, 20°C, 30°C, 40°C, and 50°C were selected.

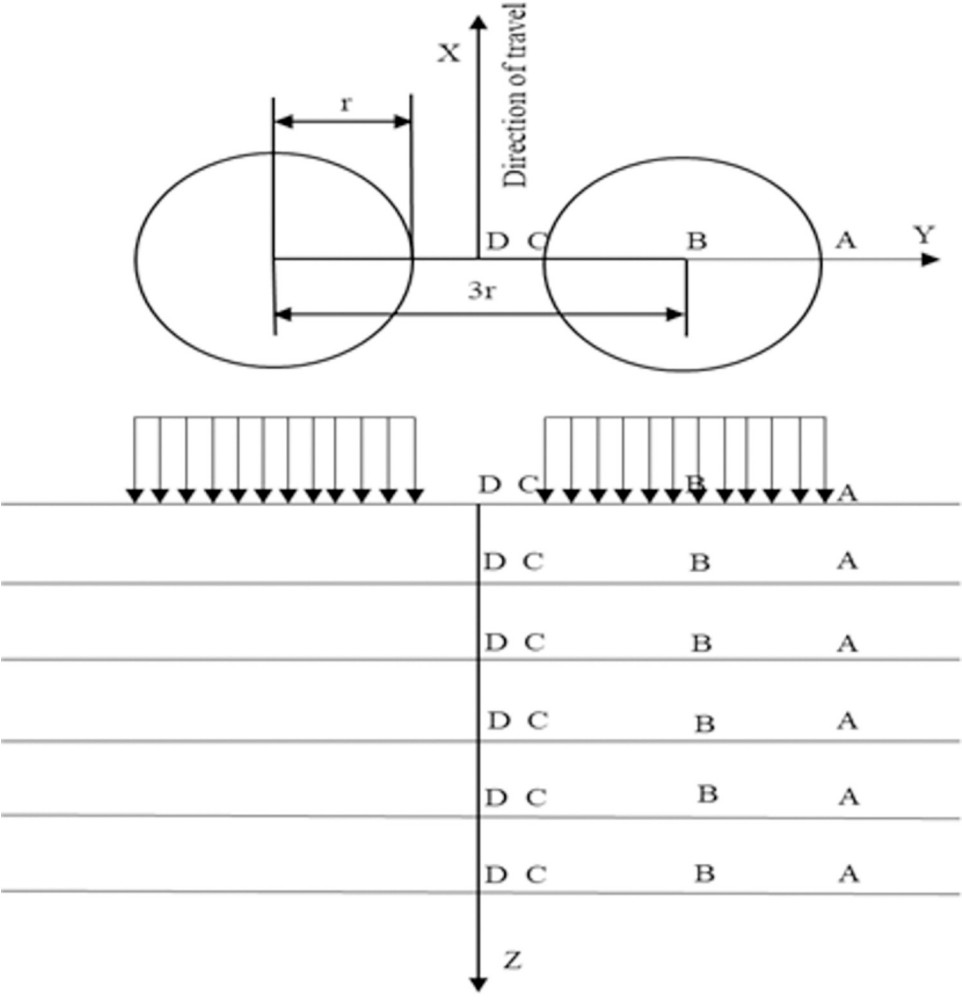

**Fig 2. Calculation point.**

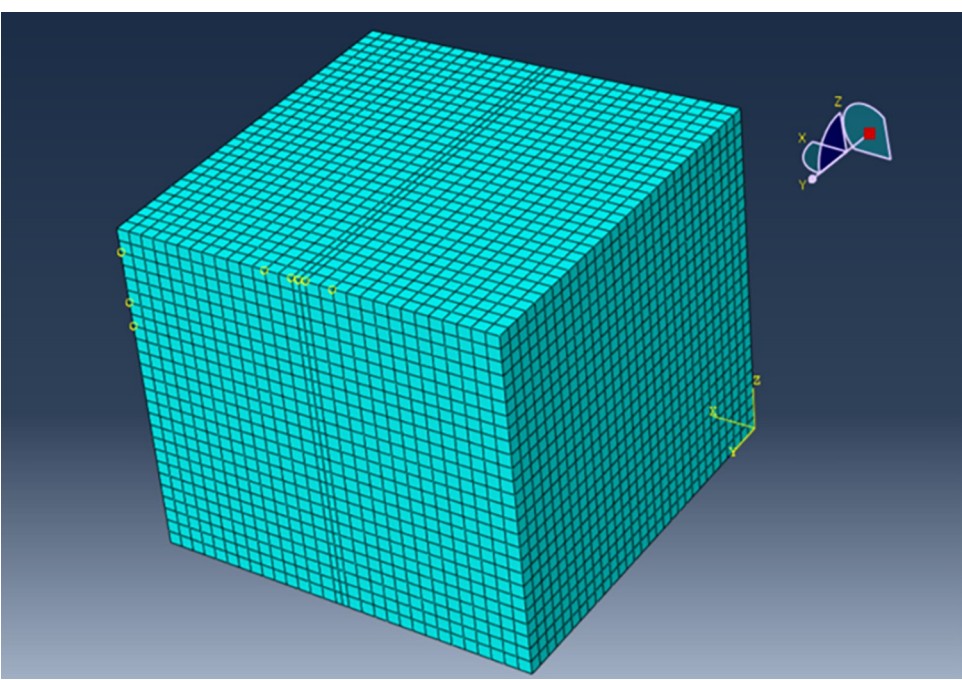

**Fig 3. Pavement structure model and grid division.**

**Fig 4. Load action position and boundary conditions.**

**Table 17. Number of repetitions of loading cycles at different load frequencies.**

| Frequency (Hz) | 25 | 10 | 5 | 1 | 0.5 | 0.1 |
|---|---|---|---|---|---|---|
| Number of Repetitions (count) | 200 | 200 | 100 | 20 | 15 | 15 |

2. Frequency: Taking into account the actual working frequency of the asphalt mixtures and the test duration, loading frequencies of 0.1 Hz, 0.5 Hz, 1 Hz, 5 Hz, 10 Hz, and 25 Hz were chosen.

3. Waveform: It is believed that the Haversine waveform corresponds to the actual vehicle load waveform experienced by the pavement. Additionally, it can be easily achieved using existing testing equipment. Therefore, the following Haversine waveform was adopted in this study.:

$$F = \frac{q}{2}(1 - cos(2\pi ft)) \tag{3}$$

In the equation, "f" represents the excitation frequency of the vehicle load, which is related to the driving speed, natural frequency of the vehicle, etc. It is a key parameter for conducting pavement mechanical response analysis.

4. Number of Loading Cycles: The number of repeated loading cycles under different load frequencies is shown in Table 17.

5. The test results are shown in Figs 5, 6.

According to Fig 5(A), it can be observed that the dynamic modulus continuously increases with the increase in frequency. This increase is more pronounced at lower temperatures, with the smallest range of dynamic modulus variation occurring at 60°C. As shown in Fig 5(B), the dynamic modulus gradually decreases with the increase in temperature, and this reduction becomes more pronounced at higher frequencies. In summary, when frequency and temperature are combined, the impact of temperature on dynamic modulus is significantly greater than the impact of frequency.

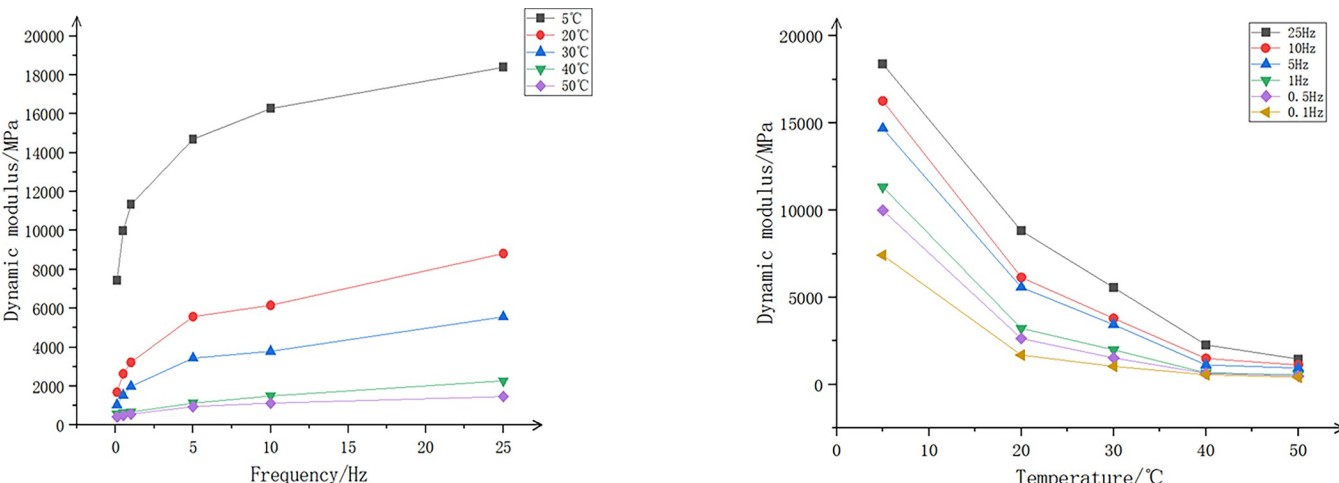

**Fig 5. Dynamic modulus curves of SMA-16 at different frequencies and temperatures.** (a)Different Temperatures, (b)Different Frequency.

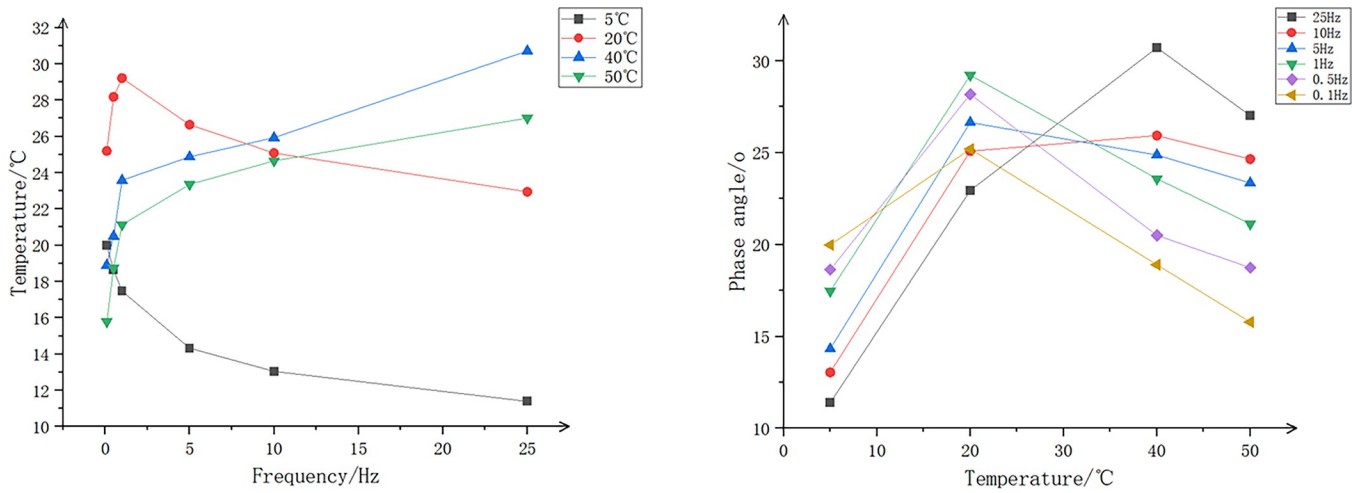

**Fig 6. Phase angle curves of SMA-16 at different frequencies and temperatures.** (a)Different Temperatures, (b)Different Frequency.

As indicated in Fig 6, at low temperatures (5°C to 20°C), the phase angle increases with rising temperature and decreases with increasing frequency. The lower the temperature, the faster the phase angle decreases with increasing frequency. At relatively higher temperatures (40°C to 50°C), the phase angle decreases with increasing temperature and increases with frequency.

## 3.2 Structural analysis of basalt fiber reinforced asphalt pavement surface

Since the upper layer of the pavement was mainly resistant to low-temperature cracking, the middle layer was mainly stable at high-temperature, and the lower layer was resistant to fatigue cracking, we used the road surface deflection, maximum shear stress, and tensile strain at the bottom of the asphalt layer as evaluation indicators to select the optimum combination of pavement structure layers.

From the dynamic modulus, the pavement surface deflection, shear stress and asphalt base tensile strain were obtained at different positions. The results of the calculations are shown in Figs 7–9 below.

Fig 7 shows that the maximum deflection of the road surface at various positions for asphalt mixtures with different dosages occurred at point B, which is the center of the single-circle load. In Fig 8, for asphalt mixtures with different dosages, the maximum shear stress at various locations was predominantly at point B, which is the single-circle load center. Only when the fiber content of the lower layer was 0.3%, the maximum shear stress at Point A was at the outer edge of the circular load. As shown in Fig 9 above, the tensile strain at the bottom of the layer varied under different dosages of asphalt mixtures at different positions. The maximum value was observed at point B, located at the center of the single circular load.

## 3.3 Orthogonal test result analysis

**3.3.1 Range analysis.** From Figs 7–9 above, the calculation results for the maximum road surface deflection, maximum shear stress and maximum tensile strain at point B were filled in Table 18, and for each index, the data of each factor level $K_{1j}$, $K_{2j}$, $K_{3j}$ and the corresponding average value $\bar{K}_{1j}$, $\bar{K}_{2j}$, $\bar{K}_{3j}$ and the range of each column $R_j$ were calculated and filled in Table 18.

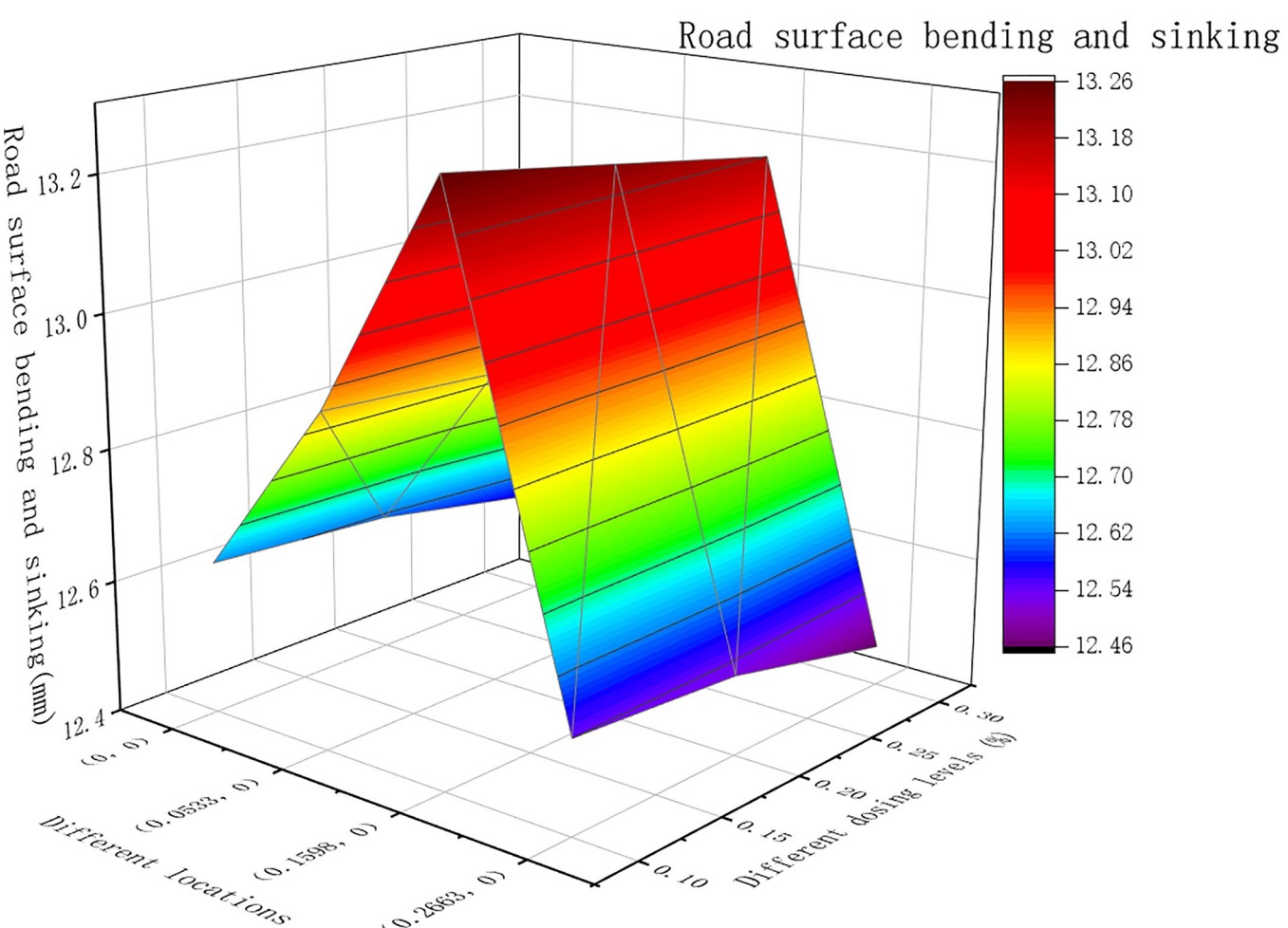

**Fig 7. The deflection of the upper layer of the road surface under different dosages.**

The unequal range $R_j$ suggests that the variations in the levels of each factor had varying effects on the experimental outcomes. Moreover, a larger range $R_j$ implies a more significant influence of the factor level changes on the experimental results. In Table 18, regarding the pavement structure composed of asphalt modified with basalt fiber, it was observed that the basalt fiber content in the middle surface layer had a greater impact on the deflection of the road surface compared to the upper and lower layers under continuous conditions. For the lower layer of basalt, the impact of the fiber content on the maximum shear stress was more significant than that of the middle surface layer and greater than that of the upper layer. The impact of the basalt fiber content in the middle surface layer on the maximum tensile strain at the bottom of the layer was more significant than that of the bottom layer and greater than that of the upper layer. If the range was smaller than that of all factors, it indicated no interaction between the factors. However, it also suggested that the aforementioned orthogonal test scheme was reasonable.

**3.3.2 Analysis of variance.** From the F distribution table, it was determined that $F_{1-0.05}$ (2, 2) = 19.00; $F_{1-0.01}$ (2, 2) = 99.00; $F_{1-0.1}$ (2, 2) = 9.00, and the road surface deflection was a characteristic of the road surface The indicators of overall stiffness and strength are shown in Table 19, $F_B = 24.00 > F_{1-0.05}(2,2) = 19.0$, $F_A = 10.00 > F_{1-0.1}(2,2) = 9.00$, $F_C < F_{1-0.1}(2, 2) = 9.00$,

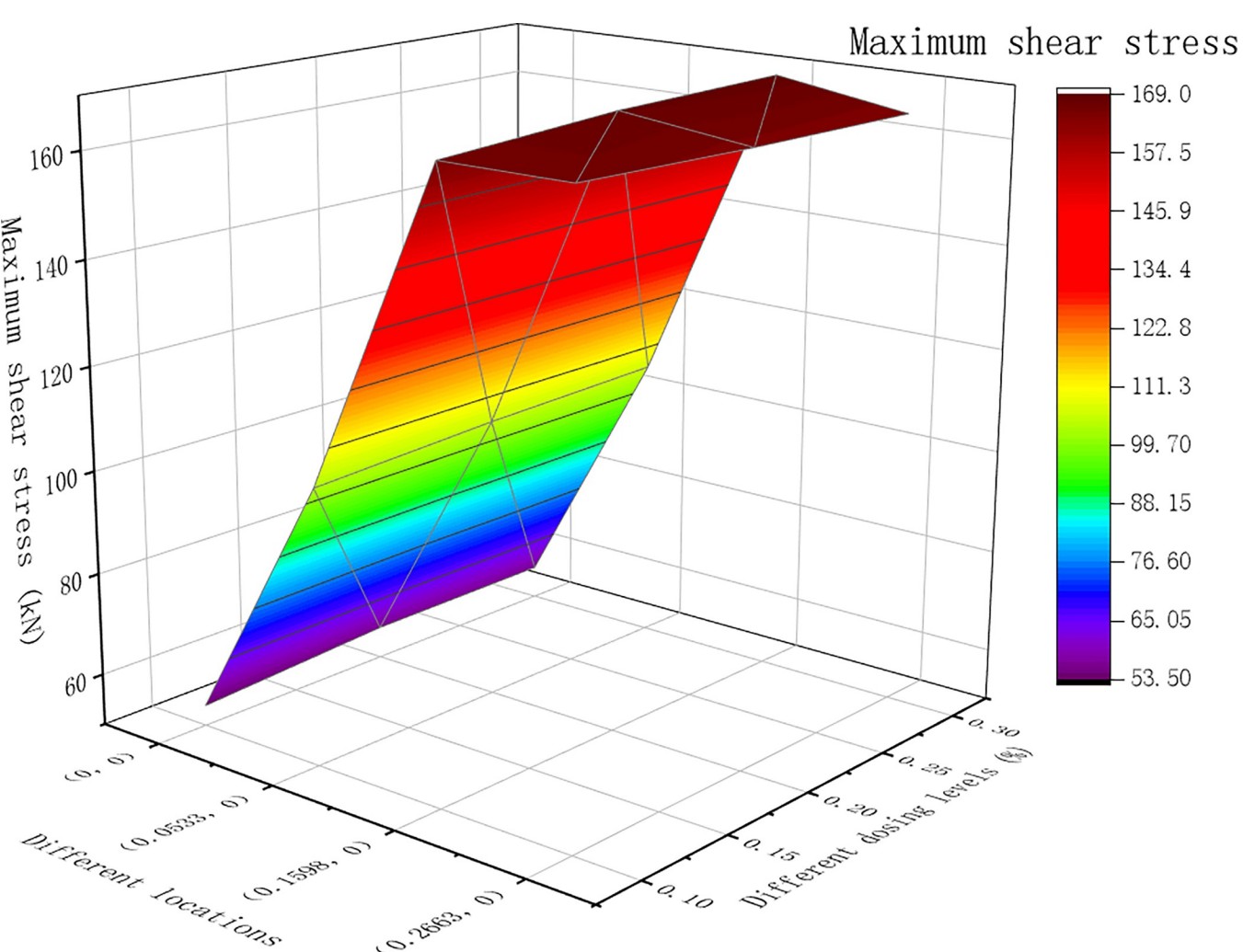

**Fig 8. The maximum shear stress of the middle surface layer under different dosages (kPa).**

factor B (basalt fiber reinforced the dense gradation asphalt concrete AC-20,BFAC-20) was significant, factor A (basalt fiber reinforced the gap-graded asphalt concrete SMA-16,BFSMA-16) was slightly significant, and factor C (basalt fiber reinforced asphalt-treated permeable base ATB-25,BFATB-25) had no effect on the test results; under the maximum shear stress index, In Table 19, $F_C = 323.51 > F_{1-0.01}(2,2) = 99.00$, $F_B = 127.61 > F_{1-0.01}(2,2) = 99.00$, and $F_A < F_{1-0.1}(2,2) = 9.00$, Factor C (BFATB-25) was highly significant, factor B (BFAC-20) was slightly significant, and factor A (BFSMA-16) had no effect on the test results. Under the maximum tensile strain index at the bottom of the layer, Table 19 shows that $F_C = 13.00 > F_{1-0.1}(2,2) = 9.00$, $F_A < F_{1-0.1}(2,2) = 9.00$, and $F_C < F_{1-0.1}(2,2) = 9.00$, which determine factor B (BFAC-20) significantly, while factor A (BFSMA-16) and factor C (BFATB-25) had no effect on the test results.

In summary, according to the results of the range analysis and variance analysis, there were two optimal solutions. Scheme 1 shows that the optimal dosage of the upper layer was 0.3% BFSMA-16, the optimal dosage of the middle surface layer was 0.1% BFAC-20, and the optimal dosage of the lower layer was 0.3% BFATB-25. The Scheme 2 was that the optimal dosage of the upper layer was 0.3% BFSMA-16, the optimal dosage of the middle surface layer was 0.3% BFAC-20, and the optimal dosage of the lower layer was 0.3% BFATB-25.

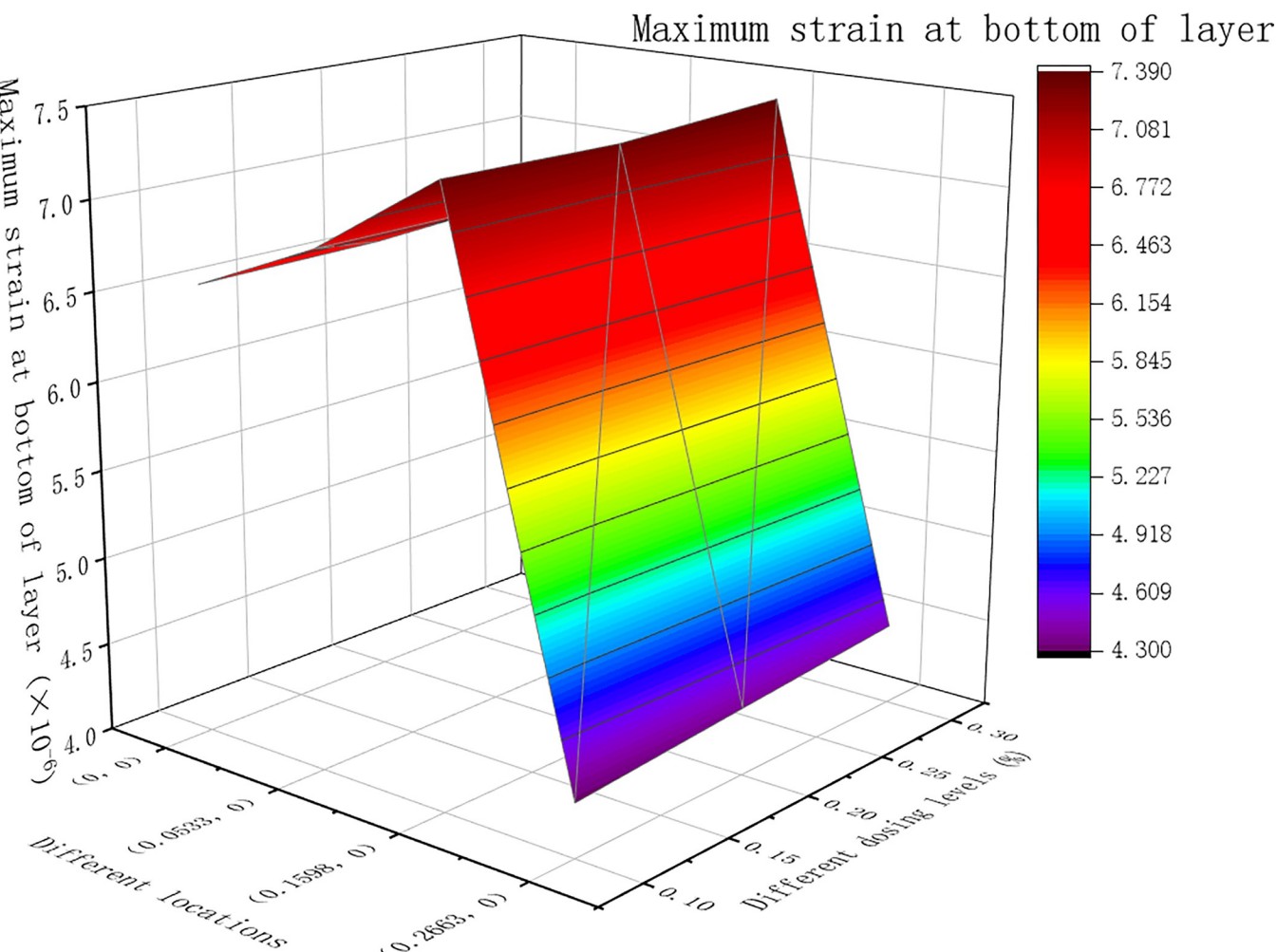

**Fig 9. The maximum tensile strain at the bottom of the layer under different dosages (×10^(-6)).**

## 3.4 Optimization analysis of basalt fiber asphalt pavement structure combination

In order to better optimize the pavement structure combination analysis, we will analyze the performance of the four pavement structures in terms of fatigue cracking life, shear stress under different axle loads, and strong base thin surface. The following Figs 10–13 shows the four pavement structures selected for this paper.

**3.4.1 Optimization analysis of pavement structure combination based on fatigue cracking life.** Three kinds of pavement structures (traditional pavement structure, scheme 1, and scheme 2 pavement structures) were selected [45, 46], and fatigue cracking calculations were conducted to determine the optimal pavement structure. The simplified model for calculating the fatigue cracking life of the asphalt mixture was:

$$N_{f1} = 2.78 \times 10^{18} \times \left(\frac{1}{\varepsilon_a}\right)^{3.97} \left(\frac{1}{E_a}\right)^{1.58} \tag{4}$$

**Table 18. Orthogonal test range analysis table.**

| Test No. | | Influencing factors | | | | Test index | | |
|---|---|---|---|---|---|---|---|---|
| | | BFSMA-16 (A) /% | BFAC-20 (B) /% | BFATB-25 (C)/ % | Blank (D) | Road surface deflection (mm) | Maximum shear stress (kPa) | The maximum tensile strain at the bottom of the layer (×10^(-6)) |
| 1 | | 0.1 | 0.1 | 0.1 | 1 | 13.44 | 168.7 | 6.984 |
| 2 | | 0.1 | 0.2 | 0.2 | 2 | 13.30 | 167.6 | 7.278 |
| 3 | | 0.1 | 0.3 | 0.3 | 3 | 13.15 | 165.3 | 7.490 |
| 4 | | 0.2 | 0.1 | 0.2 | 3 | 13.33 | 164.9 | 7.247 |
| 5 | | 0.2 | 0.2 | 0.3 | 1 | 13.16 | 163.6 | 7.413 |
| 6 | | 0.2 | 0.3 | 0.1 | 2 | 13.18 | 175.5 | 7.310 |
| 7 | | 0.3 | 0.1 | 0.3 | 2 | 13.23 | 161.6 | 7.300 |
| 8 | | 0.3 | 0.2 | 0.1 | 3 | 13.17 | 173.1 | 7.345 |
| 9 | | 0.3 | 0.3 | 0.2 | 1 | 13.15 | 171.0 | 7.511 |
| Road surface deflection | $K_{1j}$ | 39.89 | 40 | 39.79 | 39.75 | | | |
| | $K_{2j}$ | 39.67 | 39.63 | 39.78 | 39.71 | | | |
| | $K_{3j}$ | 39.55 | 39.48 | 39.54 | 39.65 | | | |
| | $\bar{K}_{1j}$ | 13.297 | 13.333 | 13.263 | 13.25 | | | |
| | $\bar{K}_{2j}$ | 13.223 | 13.21 | 13.26 | 13.237 | | | |
| | $\bar{K}_{3j}$ | 13.183 | 13.16 | 13.180 | 13.217 | | | |
| | $R_j$ | 0.114 | 0.173 | 0.083 | 0.033 | | | |
| Maximum shear stress | $K_{1j}$ | 501.6 | 495.2 | 517.3 | 503.3 | | | |
| | $K_{2j}$ | 504 | 504.3 | 503.5 | 504.7 | | | |
| | $K_{3j}$ | 505.7 | 511.8 | 490.5 | 503.3 | | | |
| | $\bar{K}_{1j}$ | 167.200 | 165.067 | 172.433 | 167.867 | | | |
| | $\bar{K}_{2j}$ | 168.000 | 168.100 | 167.833 | 168.233 | | | |
| | $\bar{K}_{3j}$ | 168.567 | 170.600 | 163.500 | 167.767 | | | |
| | $R_j$ | 1.367 | 5.533 | 8.933 | 0.466 | | | |
| Maximum tensile strain | $K_{1j}$ | 21.752 | 21.531 | 21.639 | 21.908 | | | |
| | $K_{2j}$ | 21.97 | 22.036 | 22.036 | 21.888 | | | |
| | $K_{3j}$ | 22.156 | 22.311 | 22.203 | 22.082 | | | |
| | $\bar{K}_{1j}$ | 7.251 | 7.177 | 7.213 | 7.303 | | | |
| | $\bar{K}_{2j}$ | 7.323 | 7.345 | 7.345 | 7.296 | | | |
| | $\bar{K}_{3j}$ | 7.385 | 7.437 | 7.401 | 7.361 | | | |
| | $R_j$ | 0.134 | 0.260 | 0.188 | 0.065 | | | |

$\varepsilon_a$—The tensile strain at the bottom of the asphalt mixture layer (×10^(-6)), the tensile strain at the bottom of the traditional pavement structure was 6.001, and the tensile strains at the bottom of scheme 1 and scheme 2 were 7.300 and 7.562, respectively.

$E_a$—The dynamic compression modulus (MPa) of the asphalt mixture at 20°C, the dynamic compression modulus of the traditional pavement structure was 9393 MPa, and that of scheme 1 and scheme 2 was 13124 MPa.

Through the calculation using Formula (4), the fatigue cracking life of the asphalt mixture layer of the traditional pavement structure was $N_{f0}$ = 4.2442×10^11 axles, the fatigue cracking life of the asphalt mixture layer of the pavement structure of Scheme 1 was $N_{f1}$ = 2.2973×10^11 axles, and the fatigue cracking life of the asphalt mixture layer of the pavement structure of Scheme 2 was $N_{f2}$ = 1.9972×10^11 axles.

**Table 19. Analysis of variance of influencing factors of the pavement structure.**

| | Source of variance | Influencing factors | | | | Statistical significance |
|---|---|---|---|---|---|---|
| | | Sum of squares | Degree of freedom f | Mean square sum S/f | F value | |
| **Road surface deflection (mm)** | A | 0.020 | 2 | 0.01 | 10.00 | * |
| | B | 0.048 | 2 | 0.024 | 24.00 | ** |
| | C | 0.013 | 2 | 0.0065 | 6.50 | |
| | D(e) | 0.002 | 2 | 0.001 | | |
| Maximum shear stress (kPa) | A | 2.896 | 2 | 1.448 | 8.00 | |
| | B | 46.196 | 2 | 23.098 | 127.61 | ** |
| | C | 117.109 | 2 | 58.5545 | 323.51 | *** |
| | D(e) | 0.362 | 2 | 0.181 | | |
| Maximum tensile strain at the bottom of the layer (×10^(-6)) | A | 0.027 | 2 | 0.0135 | 3.38 | |
| | B | 0.104 | 2 | 0.052 | 13.00 | ** |
| | C | 0.056 | 2 | 0.0028 | 0.70 | |
| | D(e) | 0.008 | 2 | 0.004 | | |

*** indicates highly significant, ** indicates more significant, * indicates significant.

Note: D in the source of variance is a blank column, which is the error item e.

According to the asphalt pavement structure with different fiber contents, a simplified model could be used to calculate the fatigue crack life of the asphalt mixture. For tensile strain at the base of the same asphalt mix, the smaller the moduli of the top and middle layers, the smaller the tensile strain, and the longer the fatigue life. However, in asphalt mixtures with varying fiber contents, increasing the fiber content did not enhance the fatigue life of the lower layer.

From the perspective of fiber content, the asphalt with fiber in the lower layer exhibits good fatigue performance when the fiber content is 0.3%. This also shows that the modulus value of the base asphalt mix should be controlled within a certain range instead of blindly increasing

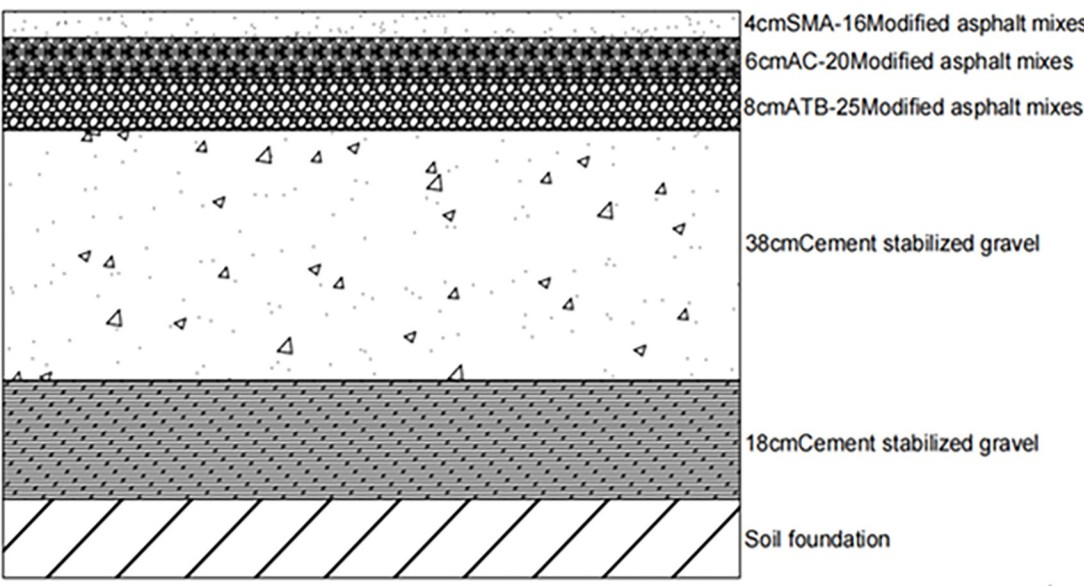

**Fig 10. Traditional pavement structure.**

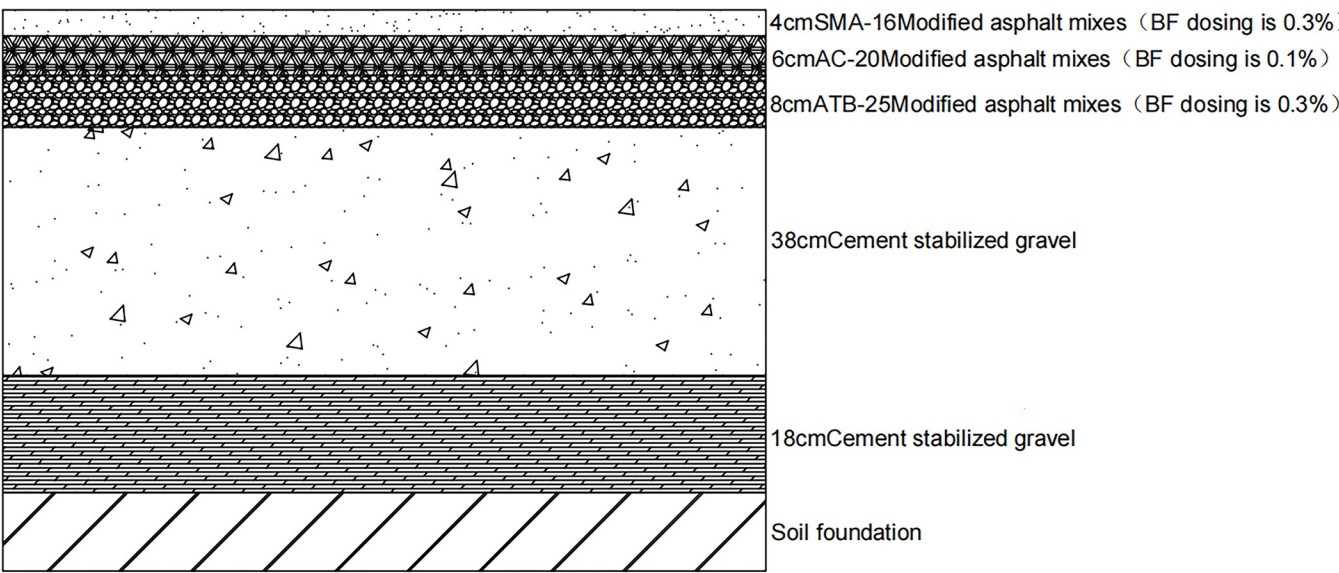

**Fig 11. Scheme 1.**

the fiber content or selecting only the best content in each layered structure to achieve the best performance of the overall structure. In addition, it is not advisable to blindly choose the method of adding the best fiber material to only the top and middle layers to improve the high- or low-temperature stability of the asphalt mix.

To enhance the other properties of the asphalt pavement by incorporating fibers, instead of focusing on improving the fatigue life, adding fibers to the lower layer would be an inappropriate choice because the fatigue life of the original asphalt pavement is already optimal. However, to increase the modulus of the upper and middle layers, it is evident that the fatigue life of the asphalt pavement would be significantly reduced if the lower layer is not adjusted accordingly.

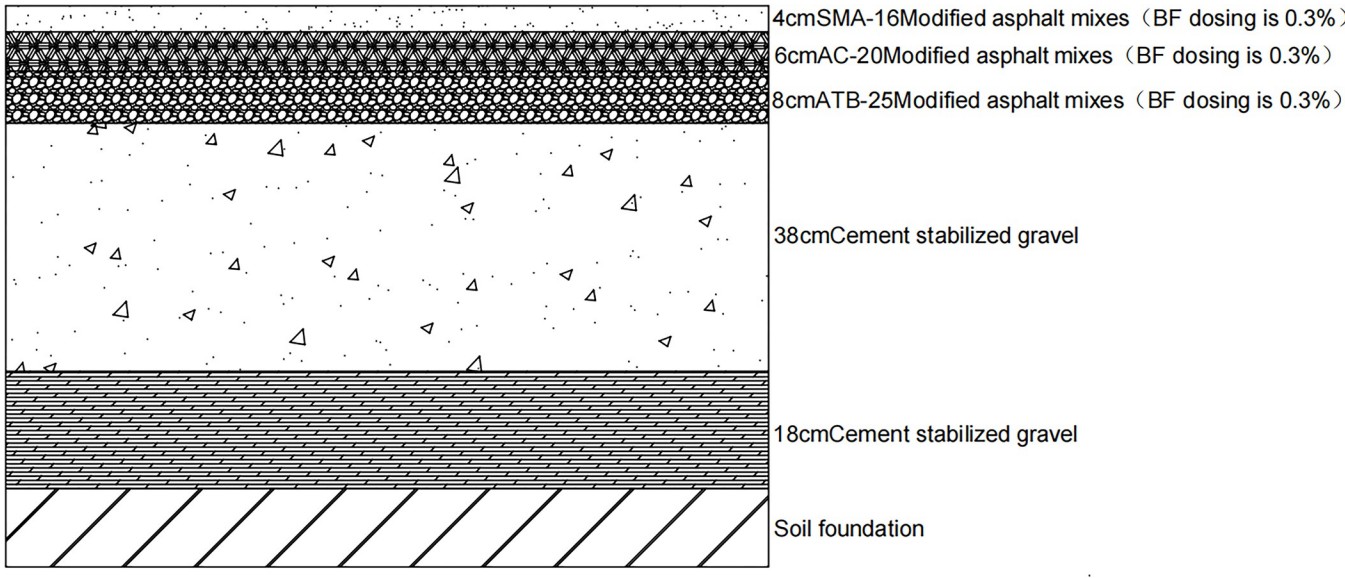

**Fig 12. Scheme 2.**

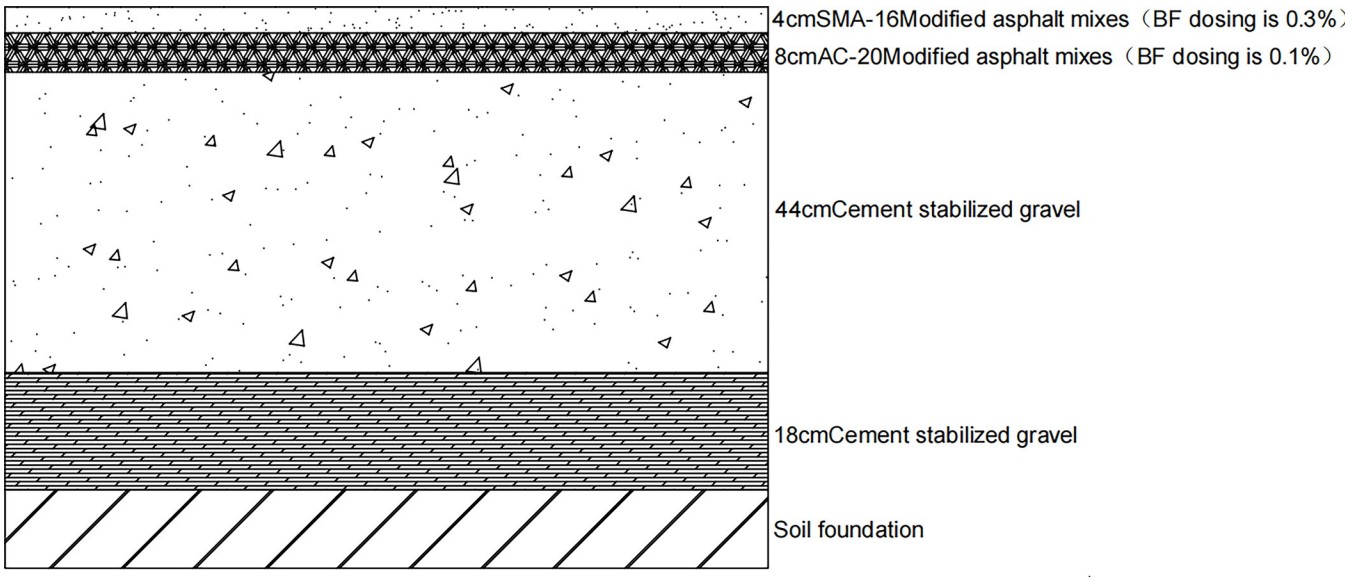

**Fig 13. Pavement structure with strong base and thin surface.**

For the 0.3% basalt fiber-reinforced asphalt mixture, under the fatigue life control index, the reasonable fiber content range for the top layer would be 0–0.3% and for the middle layer would be 0–0.1%.

Therefore, the pavement structure of the first scheme was selected. It included the optimum content of the top layer, which was 0.3% basalt fiber-reinforced gap-graded asphalt concrete SMA-16 (BFSMA-16), the optimum content of the middle layer, which was 0.1% basalt fiber-reinforced dense gradation asphalt concrete AC-20 (BFAC-20), and the optimum content of the bottom layer, which was 0.3% basalt fiber-reinforced asphalt-treated permeable base ATB-25 (BFATB-25).

**3.4.2 Optimization analysis of pavement structure based on shear stress under different axle loads.** The shear stress of the two optimal schemes was analyzed. The maximum shear stresses of the traditional pavement structure, scheme 1, and scheme 2 are shown in Table 20.

Fig 14 shows that under different axle loads, the maximum shear stress occurred at the single-circle load center. When the axle load was 100 kN, the maximum shear stress of scheme 1 was reduced by 9.4% compared to the traditional pavement structure. The maximum shear stress of the traditional pavement structure was reduced by 6.4%, while the maximum shear stress of Scheme 1 decreased by 2.8% compared to Scheme 2. When the axle load was 120 kN, the maximum shear stress of Scheme 1 was reduced by 10.5% compared to the traditional pavement structure. The maximum shear stress of scheme 2 was reduced by 7.8% compared with the traditional pavement structure, and the maximum shear stress of scheme 1 was

**Table 20. Maximum shear stress of three kinds of pavements under different axle loads.**

| Axle load P/kN | Axial pressure/MPa | Equivalent circle radius/m | Maximum shear stress/kPa | | |
|---|---|---|---|---|---|
| | | | Traditional pavement structure | Scheme 1 | Scheme 2 |
| 100 | 0.707 | 0.1065 | 176.9 | 161.60 | 166.20 |
| 120 | 0.765 | 0.1117 | 193.8 | 175.30 | 179.70 |
| 150 | 0.830 | 0.1200 | 209.4 | 187.30 | 189.60 |
| 200 | 0.906 | 0.1326 | 226.3 | 199.80 | 198.80 |

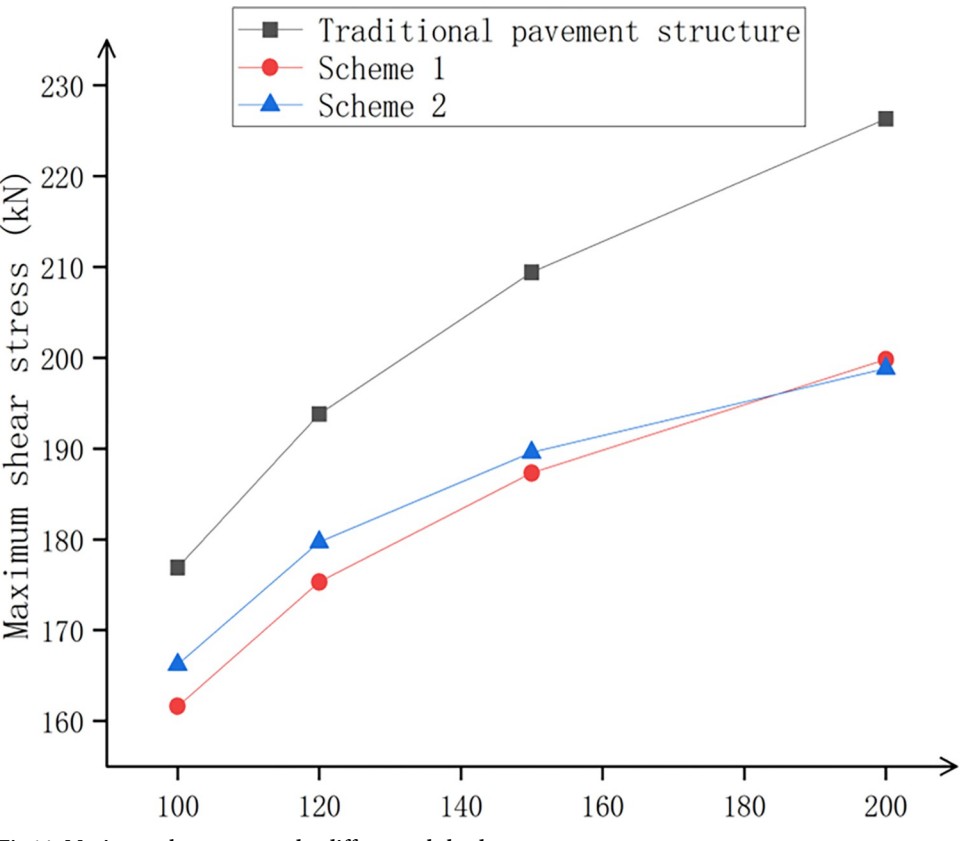

**Fig 14. Maximum shear stress under different axle loads.**

reduced by 2.5% compared with scheme 2. When the axle load was 150 kN, the maximum shear stress of scheme 1 was reduced by 11.8% compared with the traditional pavement structure, while the maximum shear stress of scheme 2 was reduced by 10.4% compared with the traditional pavement structure. Additionally, the maximum shear stress of scheme 1 was reduced by 1.2% compared with scheme 2. When the axle load was 200 kN, the maximum shear stress of scheme 1 compared to the traditional pavement structure was X. The stress was reduced by 13.2%. The maximum shear stress of scheme two decreased by 13.8% compared to the traditional pavement structure, and it was reduced by 0.5% compared to scheme one. Therefore, in the case of overloading, the optimal solution obtained from the comparison was Scheme 1 (this scheme consisted of 0.3% BFSMA-16 for the upper layer, 0.1% BFAC-20 for the middle surface layer, and 0.3% BFATB-25 for the lower layer).

**3.4.3 Optimization analysis of pavement structure combination based on the strong foundation and thin semi-rigid base.** Three kinds of pavement structures (traditional pavement structure, pavement structure with basalt fiber added, pavement structure with strong foundation and thin surface) were considered, and after calculations, the pavement surface deflection, maximum shear stress, and maximum tensile strain at the bottom of the layer are shown in Figs 15–17.

In Fig 15, compared with the traditional pavement structure, the surface deflection of the pavement structure with basalt fiber decreased by 2.2%, while the pavement structure with a strong base and thin surface increased by 0.1%. In Fig 16, compared with the traditional pavement structure, the maximum shear stress of the pavement structure with basalt fiber decreased by 6.4%, while the pavement structure with a strong base and thin surface increased

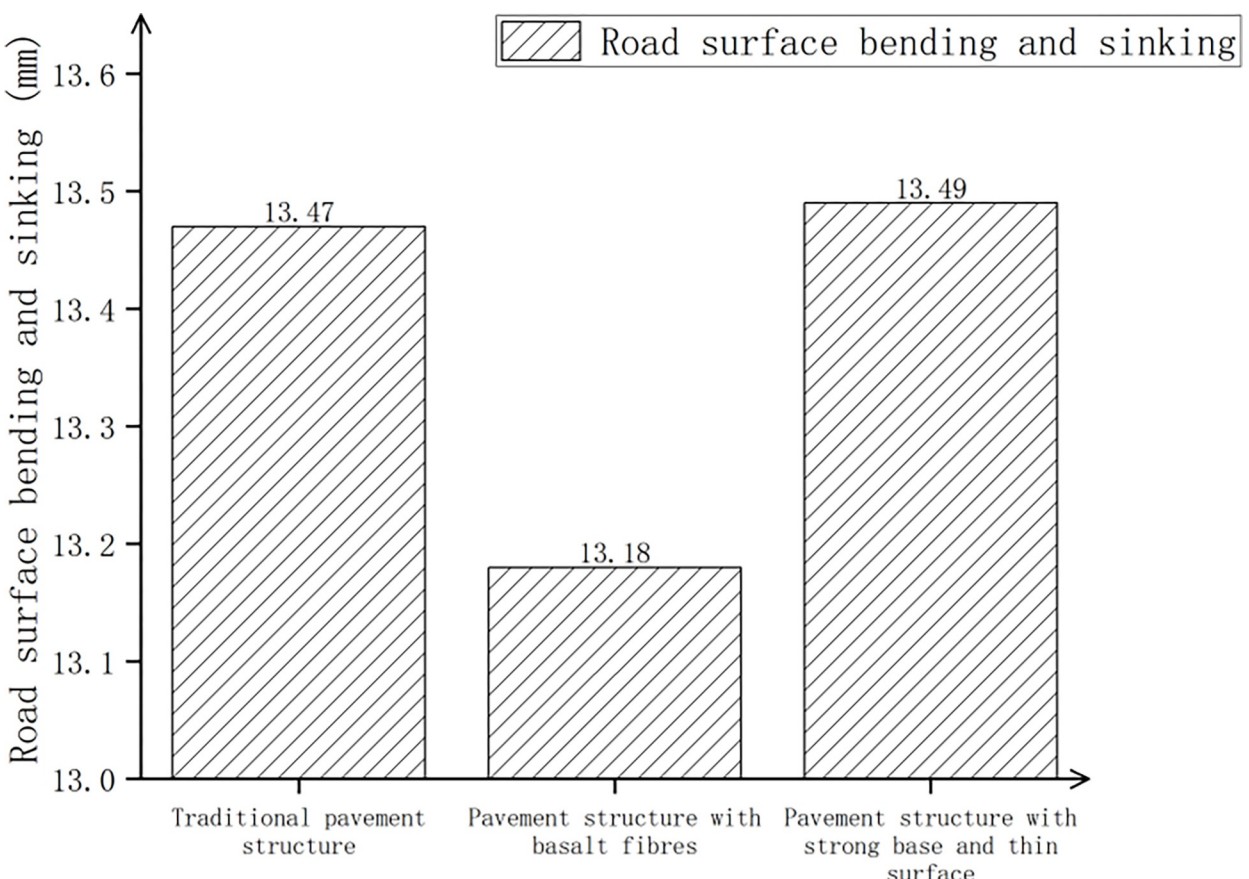

**Fig 15. Surface deflection of three pavement structures.**

by 22.4%. According to Fig 17, the maximum tensile strain at the bottom of the pavement structure increased by 21.6% when basalt fiber was added compared to the traditional pavement structure. Additionally, the maximum tensile strain at the bottom of the pavement structure increased by 9.2% when a strong base and thin surface were used. The higher the tensile strain at the bottom of the layer, the greater the strength and the material's ability to withstand large deformation without breaking. Therefore, through comprehensive consideration and analysis, we could also utilize a pavement structure comprising a sturdy base and a thin surface.

## 4. Conclusions

The use of basalt fiber in asphalt mixtures is a common practice. However, limited research has been conducted to address its impact on the mechanical properties of asphalt pavement materials, especially concerning functional requirements across various pavement layers. This project examines the influence of basalt fiber content on asphalt pavement mechanical properties using three-dimensional finite element method. The investigation addresses upper-layer resistance to low-temperature cracking, middle-layer resistance to rutting, and lower-layer resistance to fatigue cracking. The study's findings are summarized as follows:

1. With the change in basalt fiber content, the deflection of the road surface and the maximum tensile strain at the bottom of the asphalt layer of the nine asphalt pavement structures were

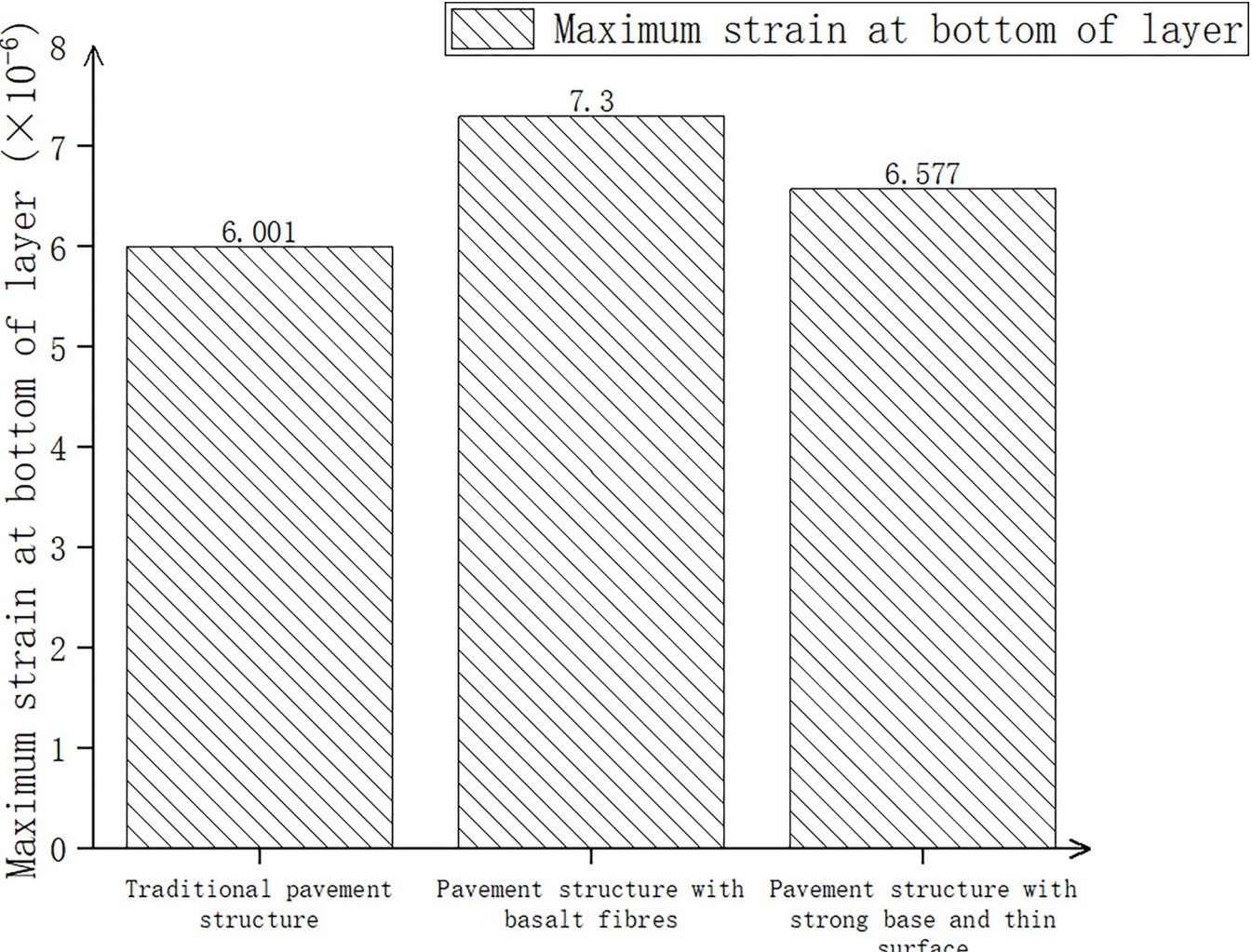

**Fig 17. Pavement deflection of three pavement structures.**

concentrated at the single-circle load center. Meanwhile, the position of the maximum shear stress was found at the single-circle load center or the single-circle load at the outer edge.

2. Through orthogonal test analysis considering fatigue life and overload, the optimal pavement structure was determined through calculation and analysis as follows: 0.3% BFS-MA-16 for the upper layer, 0.1% BFAC-20 for the middle layer, and 0.3% BFATB-25 for the lower layer. This solution can be used for the pavement surface materials of high-grade highways or the level crossing pavement surface of urban city roads.

3. Combined with the orthogonal test, it was determined that the pavement structure consisted of 4 cm BFSMA-16 (fiber content 0.3%), 8 cm BFSMA-20 (fiber content 0.1%), 44 cm cement-stabilized gravel base, and 18 cm cement-stabilized gravel subbase. The pavement surface material program is primarily designed for structural surfacing, specifically targeting long-lasting pavements currently under investigation in China.

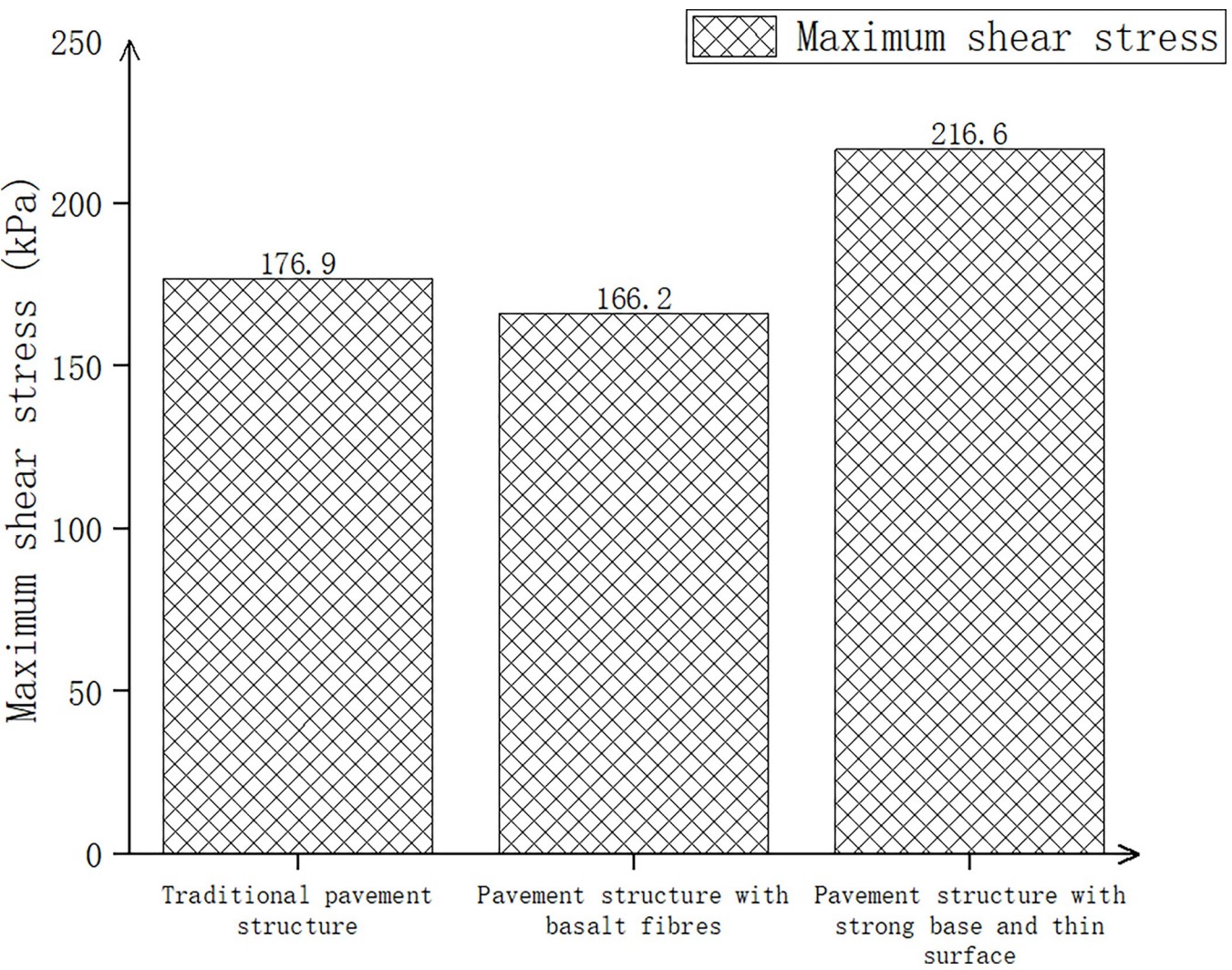

**Fig 16. Pavement deflection of three pavement structures.**

## Author Contributions

**Conceptualization:** Xiangbing Xie, Yahui He, Chenchen Liu.

**Data curation:** Yahui He.

**Formal analysis:** Xiangbing Xie, Yahui He, Kaiwei Wang.

**Funding acquisition:** Xiangbing Xie, Jinggan Shao.

**Investigation:** Xiangbing Xie, Yahui He, Kaiwei Wang.

**Methodology:** Xiangbing Xie, Kaiwei Wang, Huixia Li.

**Project administration:** Xiangbing Xie.

**Resources:** Xiangbing Xie, Yahui He.

**Software:** Yahui He, Chenchen Liu, Kaiwei Wang.

**Supervision:** Xiangbing Xie.

**Validation:** Xiangbing Xie, Zhezhe Fan.

**Visualization:** Kaiwei Wang.

**Writing – original draft:** Xiangbing Xie, Yahui He.

**Writing – review & editing:** Yahui He, Huixia Li.

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
