## [Decision Letter · Decision Letter 0]

16 Feb 2024

PONE-D-24-00063Optimization of the Dosage of Chopped Basalt Fibers in Asphalt Pavement Surface Course Materials for Semi-rigid Base with Functional RequirementsPLOS ONE

Dear Dr. Xie,

Thank you for submitting your manuscript to PLOS ONE. After careful consideration, we feel that it has merit but does not fully meet PLOS ONE’s publication criteria as it currently stands. Therefore, we invite you to submit a revised version of the manuscript that addresses the points raised during the review process.

We look forward to receiving your revised manuscript.

Kind regards,

Jiaolong Ren

Academic Editor

PLOS ONE

“The authors appreciate the financial support from the Youth Research Funds Plan of Zhengzhou University of Aeronautics (Grant No. 23HQN01007), Research on Application Technology and Equipment of Sprayed Basalt Fiber Reinforced Concrete (Grant No. 2020J-2-12), Research on Key Technologies of Application of Basalt Fiber and its Products in Highway Engineering (Grant No. 2021J5), the National Natural Science Foundation of China (Grant No. 51378474), Fund of Leading Talent in Science and Technology Innovation (Grant No. 194200510015), Science and Technology Department of Henan Province (Grant No. 192102210047), and Scientific Research and Development Project of Zhengzhou Lutong Highway Construction Co., Ltd. (Grant No. 2021JK-11).”

4. PLOS requires an ORCID iD for the corresponding author in Editorial Manager on papers submitted after December 6th, 2016. Please ensure that you have an ORCID iD and that it is validated in Editorial Manager. To do this, go to ‘Update my Information’ (in the upper left-hand corner of the main menu), and click on the Fetch/Validate link next to the ORCID field. This will take you to the ORCID site and allow you to create a new iD or authenticate a pre-existing iD in Editorial Manager. Please see the following video for instructions on linking an ORCID iD to your Editorial Manager account: https://www.youtube.com/watch?v=_xcclfuvtxQ.

5. We note that Figure 1 in your submission contain copyrighted images. All PLOS content is published under the Creative Commons Attribution License (CC BY 4.0), which means that the manuscript, images, and Supporting Information files will be freely available online, and any third party is permitted to access, download, copy, distribute, and use these materials in any way, even commercially, with proper attribution. For more information, see our copyright guidelines: http://journals.plos.org/plosone/s/licenses-and-copyright.

Reviewers' comments:

Reviewer's Responses to Questions

**Comments to the Author**

1. Is the manuscript technically sound, and do the data support the conclusions?

Reviewer #1: Yes

Reviewer #2: Partly

Reviewer #3: Partly

2. Has the statistical analysis been performed appropriately and rigorously? 

Reviewer #1: N/A

Reviewer #2: Yes

Reviewer #3: N/A

3. Have the authors made all data underlying the findings in their manuscript fully available?

Reviewer #1: Yes

Reviewer #2: No

Reviewer #3: No

4. Is the manuscript presented in an intelligible fashion and written in standard English?

Reviewer #1: Yes

Reviewer #2: Yes

Reviewer #3: No

5. Review Comments to the Author

Reviewer #1: Comments to the Author

The paper presents work with some research significance. The paper is discussing the Optimization of the Dosage of Chopped Basalt Fibers in Asphalt Pavement Surface Course Materials for Semi-rigid Base with Functional Requirements. The central argument of the paper is sound and in line with current thinking and well-focused on the topics established in PLOS ONE journal.

However, authors are encouraged to consider the following comments and include them in their paper:

- The manuscript requires English proof-reading. There are some grammatical errors and vague sentences that should be revised before publication. Authors should especially focus on revising abstract and conclusions which are the main parts of a paper.

- A comprehensive list of notations and abbreviations was not provided in the paper. The paper should be reviewed again to make sure that every single notation and abbreviation was mentioned.

- For the abstract section, it is the opinion of this reviewer to rephrase the section with valuable improvements and focusing on the main outcomes.

- For the introduction section, the section is very short and many references was mentioned. The section might rephrase and suitable refences should be mentioned.

- For the materials section, the details and properties of the materials are required for such research paper.

- For the Experimental methods section, the section is requiring more details.

- For the discussion section, it is the opinion of this reviewer to rephrase the section and add more details.

- Make sure that the bibliography style meets the journal standards.

- Nevertheless, it is the opinion of this reviewer that the research presented in the study is of a good international caliber that is relevant to the engineering community, however, it needs major revisions.

Reviewer #2: Dear All

As a result of reviewing the mentioned paper , the outcome notes are listed below, hoping it will help in evaluating and improving the study.

1. Tests for coarse aggregate used in preparing the SMA mixture are not included in the research, so please add it as it is an essential factor in preparing the mixture.

2. The mechanism for mixing and preparing the mixture according to the specification is explained accurately. If the researcher uses models from previous research belonging to him in the analysis, please mention them.

3. Preparing samples according to the specification of the mixture not included in the research paper. Please add them carefully, especially for SMA.

4. This type of mixture needs additives. Please indicate the type and percentages used

5. The analysis is inaccurate, especially the installed drawings, as they do not correspond to the reality of the situation.

Reviewer #3: While this research explores an important topic, the novelty and original contribution appear fairly limited based on existing literature. The finite element simulation methodology requires further development and more rigorous technical details to validate the approach and findings.

It would strengthen the work to re-examine the FE model in more depth. Discussing the element types, material models, boundary conditions, and calibration process in clearer terms would help establish the model's reliability and appropriate use for this application. Presenting constraint equations, visualization of results, and quantitative validation against experimental data could also make the findings more credible and applicable.

Overall, refining the FE methodology in a more specialized, analytical way that follows modeling best practices would significantly improve the professional quality and impact of this research. The author may wish to focus revisions on optimizing this technical component to clearly demonstrate the simulation capabilities and insights gained.

6. PLOS authors have the option to publish the peer review history of their article (what does this mean?). If published, this will include your full peer review and any attached files.

Reviewer #1: No

Reviewer #2: **Yes: **Rana Amir Yousif

Reviewer #3: No

---

## [Author Response · Author response to Decision Letter 0]

28 Apr 2024

Thank you very much for the editor's suggestions on my paper. I've been overhauling my posts.

---

## [Decision Letter · Decision Letter 1]

5 Jul 2024

Optimization of the Dosage of Chopped Basalt Fibers in Asphalt Pavement Surface Course Materials for Semi-rigid Base with Functional Requirements

PONE-D-24-00063R1

Dear Dr. Xie,

We’re pleased to inform you that your manuscript has been judged scientifically suitable for publication and will be formally accepted for publication once it meets all outstanding technical requirements.

Kind regards,

Jiaolong Ren

Academic Editor

PLOS ONE

Additional Editor Comments (optional):

Reviewers' comments:

Reviewer's Responses to Questions

**Comments to the Author**

1. If the authors have adequately addressed your comments raised in a previous round of review and you feel that this manuscript is now acceptable for publication, you may indicate that here to bypass the “Comments to the Author” section, enter your conflict of interest statement in the “Confidential to Editor” section, and submit your "Accept" recommendation.

Reviewer #2: All comments have been addressed

2. Is the manuscript technically sound, and do the data support the conclusions?

Reviewer #2: Yes

3. Has the statistical analysis been performed appropriately and rigorously? 

Reviewer #2: Yes

4. Have the authors made all data underlying the findings in their manuscript fully available?

Reviewer #2: Yes

5. Is the manuscript presented in an intelligible fashion and written in standard English?

Reviewer #2: Yes

6. Review Comments to the Author

Reviewer #2: PLOS authors have the option to publish the peer review history of their article (what does this mean?). If published, this will include your full peer review and any attached files.

Do you want your identity to be public for this peer review? For information about this choice, including consent withdrawal, please see our Privacy Policy.

7. PLOS authors have the option to publish the peer review history of their article (what does this mean?). If published, this will include your full peer review and any attached files.

Reviewer #2: No

---

## [Editor Report · Acceptance letter]

20 Jul 2024

PONE-D-24-00063R1 

PLOS ONE

Dear Dr. Xie, 

I'm pleased to inform you that your manuscript has been deemed suitable for publication in PLOS ONE. Congratulations! Your manuscript is now being handed over to our production team.

Kind regards, 

on behalf of

Dr. Jiaolong Ren 

Academic Editor

PLOS ONE